# POWERFUL INDEPENDENCE TESTING ON HETEROGENEOUS FEDERATED CLIENTS WITH THEORETICAL GUARANTEES

## ABSTRACT

We propose a novel federated independence testing framework that addresses both theoretical and practical challenges arising from client heterogeneity. We begin by revisiting existing federated independence testing methods and showing why they often fail to provide valid guarantees or maintain statistical power under data distributional shift across clients. Building on this analysis, we introduce a copula-based marginal alignment technique together with a stacking-based aggregation strategy that amplifies intra-client dependence while mitigating inter-client variation, yielding a theoretically sound and powerful global test. For practicality, we further accelerate the aggregation step and incorporate a privacy-preserving mechanism. On the theoretical side, we establish both the correctness of our method and the validity of the test. Extensive experiments on both synthetic and real-world datasets demonstrate the superiority of our solution over existing methods.

## 1 INTRODUCTION

Testing statistical independence is a foundational task in machine learning and statistics, supporting causal discovery (Hoyer et al., 2008; Zhang et al., 2012), representation learning (Li et al., 2021), and feature selection (Camps-Valls et al., 2010). Given the observations from a joint distribution $\mathbb{P}_{XY}$, the goal of *independence testing* (IT) is to determine whether $X$ and $Y$ are independent. As data volume expands and governance tightens, data are frequently siloed across institutions. For instance, hospitals keep their own patient records (Kidd et al., 2023), which cannot be pooled together due to privacy and security concerns, and regulation requirements. This situation motivates the federated independence testing (FedIT), which is to determine the independence relationship among variables without sharing raw data.

Compared with the well-studied *independent and identically distributed* ($i.i.d.$) setting for independence testing (Gretton et al., 2005; Zhang et al., 2012; Székely et al., 2007), FedIT is considerably more challenging because heterogeneity may degrade test power (Huang et al., 2020). To the best of our knowledge, there are only a few works that explicitly address this issue. The most recent is (Li et al., 2024), hereafter FUIT, which proposes a kernel-based federated test that accelerates computation via random features (Rahimi & Recht, 2007) and aggregates covariance-based summary statistics in the random feature space. Although FUIT achieves competitive results for federated causal discovery, we show that substantial headroom remains: its aggregation strategy is actually equivalent to naively concatenating samples in the feature space, thereby ignoring cross-client heterogeneity, lacking rigorous theoretical guarantees, and risking power loss under distribution shift. These limitations call for a theoretically grounded redesign of FedIT together with heterogeneity-aware aggregation mechanisms. Please refer to Appendix B for further review on related work.

In this paper, we propose a novel framework for federated independence testing that directly tackles both theoretical and practical challenges posed by client heterogeneity. To ensure theoretical validity and practical robustness under heterogeneous marginal distributions and dependence structures, we introduce a unified approach that combines copula-based marginal alignment with a stacking-based aggregation mechanism. The copula-based alignment exploits the key property that copulas separates marginal distributions from dependence structures, as formalized by Sklar's theorem (Sklar, 1959). This theorem ensures that any multivariate distribution can be uniquely decomposed into its

marginals and a copula that captures their dependence (Nelsen, 2006). By mapping local data into a shared copula space, our method preserves the dependence structure while mitigating discrepancies in the marginals. Complementing this, the stacking-based aggregation strategy enhances local dependence signals at each client and selectively integrates them based on their contributions to global test power. To ensure efficiency and privacy, we further design a fast and privacy-preserving aggregation protocol, making the method practical for real-world federated settings.

Our main contributions are summarized as follows: 1) We provide a systematic analysis of the challenges in FedIT under client heterogeneity and propose a novel framework that addresses both theoretical and practical challenges. 2) We introduce a copula-based marginal alignment technique combined with a stacking-based aggregation strategy that amplifies intra-client dependence while mitigating inter-client variation, resulting in a theoretically sound and powerful global test. To enhance its practicality, we further develop a fast and privacy-preserving aggregation protocol. 3) We provide theoretical guarantees on the correctness of our method and the validity of the test. 4) We empirically validate the proposed methods on both synthetic and real-world benchmarks, demonstrating their practical effectiveness, and superiority over existing methods.

## 2 PRELIMINARIES AND PROBLEM FORMULATION

We begin by recalling the classical hypothesis-testing framework for statistical independence, and then formalize the FedIT setting with heterogeneous clients.

**The hypothesis testing framework.** The goal of independence testing is to decide whether two random variables $X$ and $Y$ are independent ($X \perp\!\!\!\perp Y$). Formally,

$$\mathcal{H}_0 : \mathbb{P}_{XY} = \mathbb{P}_X \mathbb{P}_Y \text{ versus } \mathcal{H}_1 : \mathbb{P}_{XY} \neq \mathbb{P}_X \mathbb{P}_Y. \tag{1}$$

The testing procedure is as follows: First, define the statistic $\rho$ and calculate its estimated value using the samples. Then, choose a significance level $\alpha$ (typically set to 0.05), which represents the probability that the sampling of $\rho$ under $\mathcal{H}_0$ is at least as extreme as the observed value. Finally, the null hypothesis $\mathcal{H}_0$ is rejected if the $p$-value is not greater than $\alpha$.

In this procedure, two types of errors may occur. Type I error occurs when $\mathcal{H}_0$ is falsely rejected, while Type II error happens when $\mathcal{H}_0$ is incorrect but not rejected. A good test (Zhang et al., 2012) needs to control Type I error within $\alpha$ while maximizing the testing power ($1 -$ Type II error rate).

**The federated setting with heterogeneous clients.** We consider a federated setting with $K$ clients (distinct domains). Client $k \in [K]$ [1] holds $n_k$ samples $\boldsymbol{Z}_k = \{(x_i^{(k)}, y_i^{(k)})\}_{i=1}^{n_k}$ drawn from a joint distribution $\mathbb{P}_{X_k Y_k}$ on $X_k \in \mathbb{R}^{d_x}$ and $Y_k \in \mathbb{R}^{d_y}$. Let $\mathbb{P}_{X_k}$ and $\mathbb{P}_{Y_k}$ denote the corresponding marginals. All samples are mutually independent both intra- and inter-client. In federated independence testing and causal discovery (Huang et al., 2020; Li et al., 2024), it is common to assume that, although data distributions may vary by client, the dependence relationship between the variables is consistent.

**Assumption 1** (Consistent dependence assumption). *We assume that the dependence relationship between $X_k$ and $Y_k$ is consistent across clients. That is, either all clients satisfy independence ($X_k \perp\!\!\!\perp Y_k$) or all exhibit dependence ($X_k \not\perp\!\!\!\perp Y_k$).*

This assumption reflects many real-world federated applications (e.g., multi-hospital studies, cross-region deployments), where a common data-generating mechanism governs all domains, even as local conditions differ without flipping the underlying dependence status.

**Assumption 2** (Heterogeneous clients assumption). *The dependence mechanism (e.g., strength or functional relationship) and the marginal distributions $\mathbb{P}_{X_k}$ and $\mathbb{P}_{Y_k}$ may vary across clients.*

Together, these assumptions define a realistic yet challenging regime: the global dependence status is common, but local distributions are heterogeneous. Our goal is to design a test that aggregates cross-client evidence to infer the shared dependence status while handling client heterogeneity.

---

[1] Throughout, the symbol $[m]$ denotes the set $\{1, 2, ..., m\}$.

## 3  LIMITATIONS OF EXISTING FEDERATED INDEPENDENCE TESTS

In this section, we begin by identifying the aggregation challenges of federated independence testing (FedIT) under client heterogeneity and then show that existing methods not only face fundamental theoretical limitations but can also suffer substantial power loss in realistic scenarios.

### 3.1  CHALLENGES OF FEDERATED AGGREGATION UNDER CLIENT HETEROGENEITY

Fig. 1 summarizes two key challenges. On the left, we illustrate the pitfall of naive aggregation strategies caused by heterogeneous marginal distributions. Two failure modes can occur: (i) independence relationships within individual clients lead to spurious dependence after aggregation; (ii) dependence relationships within individual clients lead to spurious independence after aggregation.

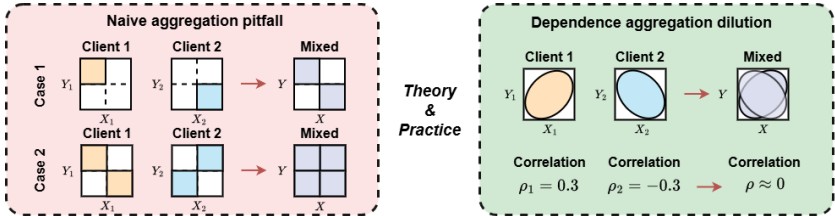

Figure 1: (Left) Naive aggregation pitfall: local independence may appear dependent after aggregation, or local dependence may cancel and appear independent. (Right) Dependence dilution: strong local dependencies are weakened after aggregation due to client heterogeneity.

On the right, we highlight a practical challenge stemming from heterogeneous functional relationships across clients. Consider a simple example: Client 1's variables $(X_1, Y_1)$ follow a bivariate Gaussian distribution with a correlation coefficient of 0.3, whereas Client 2's variables $(X_2, Y_2)$ follow a bivariate Gaussian with a correlation coefficient of $-0.3$. When combined, these opposing correlations cancel out, producing an aggregated relationship that appears uncorrelated. This scenario poses a major difficulty for independence testing, as it requires detecting higher-order dependencies beyond linear correlation, which in turn reduces the statistical power of the test.

Building on this, we later revisit existing methods and show that they are incapable of addressing either the theoretical aggregation pitfall or the practical signal dilution challenge described above.

### 3.2  REVISITING EXISTING FEDERATED INDEPENDENCE TESTING METHODS

We focus on a representative class of FedIT methods (Li et al., 2024) that extend the Kernel-based Independence Test (KIT) (Zhang et al., 2012) to the federated setting, which we refer to as FUIT. In the following, we show that FUIT's aggregation strategy is essentially equivalent to naively concatenating client samples in the feature space; consequently, it cannot overcome the limitations before.

Formally, let $\boldsymbol{x} = (\boldsymbol{x}_1, \boldsymbol{x}_2, \ldots, \boldsymbol{x}_n) \in \mathbb{R}^{d_x \times n}$ denote $n$ samples of dimension $d_x$, and similarly let $\boldsymbol{y} \in \mathbb{R}^{d_y \times n}$. The statistic of KIT is defined as $T = n\|C_{xy}\|_F^2$, where the covariance matrix is $C_{xy} = \frac{1}{n}\tilde{\phi}(\boldsymbol{x})^T \tilde{\phi}(\boldsymbol{y}) \in \mathbb{R}^{h \times h}$. Here, $\tilde{\phi}(\boldsymbol{x}) \in \mathbb{R}^{n \times h}$ is the centered random feature matrix, given by $\tilde{\phi}(\boldsymbol{x}) := \mathbf{H}\phi(\boldsymbol{x})$, $\mathbf{H} = \mathbf{I} - \frac{1}{n}\mathbf{1}\mathbf{1}^T$ where $\mathbf{1} \in \mathbb{R}^{n \times 1}$ is the all-one vector. The feature map is $\phi(\boldsymbol{x}) := \sqrt{2/h}\left[\cos(w_1^T \boldsymbol{x} + b_1); \ldots; \cos(w_h^T \boldsymbol{x} + b_h)\right]^T \in \mathbb{R}^{n \times h}$ with $w_i \sim \mathbb{P}(w)$ and $b_i \sim \text{Uniform}(0, 2\pi)$. Here, $h$ denotes the number of random features. The same construction applies to $\phi(\boldsymbol{y})$. To determine the rejection threshold, a two-parameter Gamma distribution is used to approximate the null distribution under $\mathcal{H}_0$, with parameters determined by the first two moments:

$$\mathcal{E}_0 := \mathbb{E}_{\mathcal{H}_0}[T] = \text{Tr}(C_{xx}) \cdot \text{Tr}(C_{yy}), \quad \mathcal{V}_0 := \text{Var}_{\mathcal{H}_0}[T] = 2\|C_{xx}\|_F^2 \cdot \|C_{yy}\|_F^2, \quad (2)$$

where $C_{xx}$ and $C_{yy}$ are defined analogously as $C_{xy}$. Then, the critical threshold $\widehat{c_\alpha}$ can be obtained:

$$\mathcal{H}_0 : n\|C_{xy}\|_F^2 \sim \frac{x^{\gamma-1}e^{-x/\beta}}{\beta^\gamma \Gamma(\gamma)}, \gamma = \frac{\mathcal{E}_0^2}{\mathcal{V}_0}, \beta = \frac{\mathcal{V}_0}{\mathcal{E}_0}, \quad s.t. \int_0^{\frac{\widehat{c_\alpha}}{\beta}} \frac{x^{\gamma-1}e^{-x}}{\Gamma(\gamma)}dx = 1 - \alpha, \quad (3)$$

where $\Gamma(\cdot)$ is the Gamma function. Finally, independence is decided by comparing $T$ with $\widehat{c_\alpha}$.

For FUIT, the key distinction from KIT lies in the aggregation process that its cross-covariance matrix is computed via client-wise aggregation, $C_{xy} = \sum_k C_{xy}^{(k)}$, $n = \sum_k n_k$, where $C_{xy}^{(k)}$ is calculated by $C_{xy}^{(k)} = \frac{1}{n} \phi(\boldsymbol{x}^{(k)})^T \mathbf{H}_k^T \mathbf{H}_k \phi(\boldsymbol{y}^{(k)})$, and $\phi(\boldsymbol{x}^{(k)}) \in \mathbb{R}^{n_k \times h}$ is obtained by replacing $\boldsymbol{x}$ with the local sample vector $\boldsymbol{x}^{(k)} := (x_{k;1}, x_{k;1}, ..., x_{k,n_k}) \in \mathbb{R}^{d_x \times n_k}$ for client $k$. The centering matrix is $\mathbf{H}_k = \mathbf{I} - \frac{1}{n} \mathbf{1}_{n_k} \mathbf{1}_{n_k}^T$. The terms $\phi(\boldsymbol{y}^{(k)})$, $C_{xx}^{(k)}$ and $C_{yy}^{(k)}$ are defined by analogy.

We now show that this aggregation is equivalent to applying the method to the simple concatenation of client features. Define $f_k := \mathbf{H}_k \phi(\boldsymbol{x}^{(k)}) \in \mathbb{R}^{n_k \times h}$, $f_{con} := [f_1; f_2; ...; f_K] \in \mathbb{R}^{n \times h}$. It follows that $C_{xy} = \sum_k C_{xy}^{(k)} = \frac{1}{n} \sum_k f_k^T f_k = \frac{1}{n} f_{con}^T f_{con}$, and note that the term $\frac{1}{n} f_{con}^T f_{con}$ is exactly the same as computing the statistic on the fully concatenated features from all clients.

As discussed earlier, such naive aggregation is problematic in the presence of heterogeneity. As a result, it can lead to uncontrolled Type I error or severely reduced test power, particularly when client-specific marginals or dependence structures differ. We therefore conclude that existing FedIT methods are inadequate for addressing the fundamental challenges posed by heterogeneous clients. In the next section, we introduce a new approach designed specifically to handle these limitations.

# 4 METHODS

In this section, we present our solutions to the previously analyzed challenges. The overall framework is illustrated in Fig. 2, which outlines the key steps. The core module consists of a copula transform to achieve marginal alignment, together with a Canonical Correlation Analysis (CCA, Härdle & Simar (2007)) with random projections to strengthen intra-client dependencies. These intra-client procedures, in turn, enable a more effective aggregation process, thereby enhancing the power of the global test. In what follows, we introduce each component of the framework in detail.

## 4.1 THE COPULA OF DISTRIBUTIONS

The copula (Nelsen, 2006) plays a crucial role when studying the dependence among random variables. The copula contains all the information needed to measure dependence, and it is invariant to any nonlinear strictly increasing transformations of the marginal variables.

**Definition 1** (Copula transformation (Sklar, 1959)). *Consider a $d$-dimensional random vector $\boldsymbol{X} = (X_1, ..., X_d)$ with continuous marginal cumulative distribution functions (cdfs) $F_i$, $i \in [d]$. Then, the joint cumulative distribution of $\boldsymbol{X}$ is uniquely expressed as:*

$$\boldsymbol{F}(\boldsymbol{X}) := \boldsymbol{F}(X_1, ..., X_d) = \boldsymbol{C}(F_1(X_1), ..., F_d(X_d)), \tag{4}$$

*where the distribution $\boldsymbol{C}$ is known as the copula of $\boldsymbol{X}$.*

The copula has uniform marginals as shown in the following theorem:

**Theorem 1** (Probability integral transform (Nelsen, 2006)). *For a random variable $X$ with cdf $F$, the random variable $U = F(X)$ is uniformly distributed on $[0, 1]$.*

Thus, the copula $\boldsymbol{C}$ has $d$ marginals $U_i = F_i(X_i) \sim \text{Uniform}[0, 1]$, $i \in [d]$. Given a sample matrix $[x_{j,i}]_{n \times d}$ with $n$ samples, we can estimate $F_i$ using the empirical marginal cdf defined as $F_{n,i}(x) := \frac{1}{n} \sum_{j=1}^n \mathbb{I}[x_{j,i} \leq x]$, $i \in [d]$, where $\mathbb{I}$ is the indicator function. Then, for a $d$-dimensional vector $\boldsymbol{x}$, the empirical copula transformation is $\boldsymbol{F}_n(\boldsymbol{x}) := [F_{n,1}(x_1), F_{n,2}(x_2), ..., F_{n,d}(x_d)]$, which converges to the true transformation as the sample size increases:

**Theorem 2** (Convergence of the empirical copula Póczos et al. (2012)). *Let $\boldsymbol{F}$ be the copula transformation defined above, and $\boldsymbol{F}_n$ be the corresponding empirical copula transformation, then*

$$\mathbb{P}\left[\sup_{\boldsymbol{x} \in \mathbb{R}^d} \|\boldsymbol{F}(\boldsymbol{x}) - \boldsymbol{F}_n(\boldsymbol{x})\|_2 > \epsilon\right] \leq 2d \exp\left(-\frac{2n\epsilon^2}{d}\right). \tag{5}$$

The speed of exponential convergence with respect to the sample size guarantees the performance of the copula transform method in practical applications. Calculating $\boldsymbol{F}_n(\boldsymbol{X})$ involves sorting the marginals of $\boldsymbol{X} \in \mathbb{R}^d$ with $n$ samples, thus $\mathcal{O}(dn \log n)$ operations.

Figure 2: An overview of the process of our FedIT-CS framework. Each client first applies an empirical copula transformation to ensure translation and scale invariance, followed by random feature projection to capture the nonlinear dependency signal. The resulting features are combined linearly to maximize dependence strength. Then, to facilitate aggregation across heterogeneous clients, a second copula transformation aligns marginal distributions. A privacy-preserving module is adopted before aggregation, and a subset of clients are selected to maximize the overall test power.

## 4.2 INTRA-CLIENT DEPENDENCE MEASURE VIA RANDOM PROJECTIONS

The goal of this section is to extract the intra-client dependence signal, thereby enabling a more effective aggregation procedure. We build upon the Hirschfeld–Gebelein–Rényi Maximum Correlation Coefficient (HGR) (Gebelein, 1941), which defines dependence as

$$\text{hgr}(X, Y) = \sup_{f,g} \rho(f(X), g(Y)), \tag{6}$$

where the supremum is taken over all Borel-measurable functions $f$ and $g$ with finite variance, and $\rho$ denotes Pearson's correlation coefficient. Intuitively, HGR captures the maximal correlation attainable under nonlinear transformations of the variables. In practice, a common class of estimators approximates the transformation functions $f$ and $g$ using random projections, as proposed by (Lopez-Paz et al., 2013). This approach bypasses the need for an explicit optimization step, leveraging the properties of random projections as outlined below.

**Theorem 3** (Approximation with random projections Póczos et al. (2012)). *Let $p(w)$ be a distribution on $\Omega$ and $\sup_{x,w} |\phi(x; w)| \leq 1$. Let $\mathcal{F} = \{f(x) = \int_{\Omega} u(w)\phi(x; w)dw \mid |u(w)| \leq Cp(w)\}$. Draw $w_1, ..., w_h$ i.i.d from $p(w)$. Further let $\delta > 0$, and $c$ be some L-Lipschitz loss function, and consider data $\{x_i, o_i\}_{i=1}^n$ drawn i.i.d from some arbitrary $\mathbb{P}_{XO}$. The linear regression coefficient $u_1, ..., u_h$ for which $f_h(x) = \sum_{i=1}^h u_i\phi(x; w_i)$ minimizes the empirical risk $c(f_h(x), o)$ has a distance from the c-optimal estimator in $\mathcal{F}$ bounded by*

$$\mathbb{E}_{\mathbb{P}}[c(f_h(x), o)] - \min_{f \in \mathcal{F}} \mathbb{E}_{\mathbb{P}}[c(f(x), o)] \leq O\left(\left(\frac{1}{\sqrt{n}} + \frac{1}{\sqrt{h}}\right) LC\sqrt{\log\frac{1}{\delta}}\right) \tag{7}$$

*with probability at least $1 - 2\delta$.*

Intuitively, randomly selecting $\{w_i\}_{i=1}^n$ instead of optimizing them causes only bounded error. Therefore, Eq. (6) can be approximated as optimizing $\boldsymbol{u}$ and $\boldsymbol{v}$ to maximize $\rho(\boldsymbol{u}^T\phi(\boldsymbol{x}), \boldsymbol{v}^T\phi(\boldsymbol{y}))$, where $\phi(\boldsymbol{x})$ and $\phi(\boldsymbol{y})$ are random projections of $\boldsymbol{x}$ and $\boldsymbol{y}$. The remaining task is to choose the nonlinear projection functions and to optimize $\boldsymbol{u}$ and $\boldsymbol{v}$. Following the choice of Lopez-Paz et al. (2013), we choose cosine projections, which are also called random Fourier feature (RFF) (Rahimi & Recht, 2007). Formally, the weights are drawn as $w_i \sim \mathcal{N}(0, s\mathbf{I}), b_i \sim \text{Uniform}(-\pi, \pi)$ for $i \in [h]$, where $s$ is a tunable parameter, typically set empirically as a linear function of the input dimensionality. The corresponding RFF is then defined by $\phi(\boldsymbol{x}) := \left[\cos(w_1^T\boldsymbol{x} + b_1); \ldots; \cos(w_h^T\boldsymbol{x} + b_h)\right]^T \in \mathbb{R}^{n \times h}$, and analogously for $\phi(\boldsymbol{y})$. The task of finding $\boldsymbol{u}$ and $\boldsymbol{v}$ then turns to a Canonical Correlation Analysis (CCA) problem (Härdle & Simar, 2007). Formally, let $C_{xy} := \text{cov}(\phi(\boldsymbol{x}), \phi(\boldsymbol{y})) \in \mathbb{R}^{h \times h}$ and define $C_{xx}$ and $C_{yy}$ analogously. The problem is thus equivalent to solving the eigenproblem:

$$\begin{bmatrix} 0 & C_{xx}^{-1}C_{xy} \\ C_{yy}^{-1}C_{yx} & 0 \end{bmatrix} \begin{bmatrix} \boldsymbol{u} \\ \boldsymbol{v} \end{bmatrix} = \rho^2 \begin{bmatrix} \boldsymbol{u} \\ \boldsymbol{v} \end{bmatrix}, \tag{8}$$

where the largest eigenvalue corresponds to the leading canonical correlation $\rho_1$. As a result, after applying the nonlinear projections and identifying the canonical directions $\boldsymbol{u}$ and $\boldsymbol{v}$, the nonlinear components of the dependence structure are effectively captured. This, in turn, facilitates the subsequent aggregation process. Specifically, the outputs of this procedure are 1-dimensional random variables $\boldsymbol{u}^T\phi(\boldsymbol{x}) \in \mathbb{R}^{1 \times n}$ and $\boldsymbol{v}^T\phi(\boldsymbol{y}) \in \mathbb{R}^{1 \times n}$ with the correlation $\rho_1$.

**Remark.** This method is closely related to KIT in Sec. 3.2, as both rely on $C_{xy}$. The key difference is that KIT considers all eigenvalues (via the Frobenius norm), while our approach focuses on the largest one (through CCA). In federated settings, this helps avoid eigenvalue cancellation and dependence dilution. For instance, in Sec. 3.2, $\Sigma_{1,xy} = 0.3$ and $\Sigma_{2,xy} = -0.3$ sum to zero, but according to Eq. (8), our method still retains the largest eigenvalue of 0.3 for both clients, providing consistent dependence signals. Moreover, the resulting low-dimensional output reduces communication cost and enhances privacy, as it makes the reconstruction of a client's raw data more difficult.

For subsequent aggregation, we again use the copula method to align the marginal distributions of $\boldsymbol{u}^T\phi(\boldsymbol{x})$ and $\boldsymbol{v}^T\phi(\boldsymbol{y})$. The resulting quantities are denoted as $(\boldsymbol{r_x}, \boldsymbol{r_y})$, where $\boldsymbol{r_x}, \boldsymbol{r_y} \in \mathbb{R}^{1 \times n}$, with the corresponding random variables $R_x$ and $R_y$, whose marginal distributions are given by $R_x, R_y \sim \text{Uniform}(0, 1)$. Next, we introduce the detailed process of aggregation.

### 4.3 STACKING-BASED AGGREGATION STRATEGY

For each client $k \in [K]$, after the intra-client dependence modeling process, we obtain the output copula samples $(\boldsymbol{r_x}^{(k)}, \boldsymbol{r_y}^{(k)})$. We then proceed to the aggregation process. For ease of exposition, we first ignore the client subset selection process (which will be introduced later) and focus on how to compute the sub-statistics transmitted to the server, as well as the server-side summary procedure.

**Computation of summary statistics.** Since the nonlinear dependencies have already been captured during the intra-client stage, the inter-client aggregation only needs to summarize linear correlations. Let the selected subset of client ids be denoted as $\mathcal{I}$, hence the total sample size is $n_{\mathcal{I}} = \sum_{k \in \mathcal{I}} n_k$. The global correlation coefficient can then be fully computed from the second-order moments statistics, which are obtained locally at each client. Formally, client $k$ computes:

$$e_x^{(k)} = \sum_{i=1}^{n_k} r_{x;i}^{(k)}, \quad m_{xy}^{(k)} = \sum_{i=1}^{n_k} r_{x;i}^{(k)}r_{y;i}^{(k)}, \quad m_{xx}^{(k)} = \sum_{i=1}^{n_k} r_{x;i}^{(k)}r_{x;i}^{(k)}, \quad m_{yy}^{(k)} = \sum_{i=1}^{n_k} r_{y;i}^{(k)}r_{y;i}^{(k)}. \quad (9)$$

The server then aggregates these local statistics to obtain:

$$e_x^{\mathcal{I}} = \sum_{k \in \mathcal{I}} e_x^{(k)}, \quad m_{xy}^{\mathcal{I}} = \sum_{k \in \mathcal{I}} m_{xy}^{(k)}, \quad \rho_{xy}^{\mathcal{I}} = \frac{n_{\mathcal{I}} m_{xy}^{\mathcal{I}} - e_x^{\mathcal{I}} e_y^{\mathcal{I}}}{\sqrt{n_{\mathcal{I}} m_{xx}^{\mathcal{I}} - (e_x^{\mathcal{I}})^2}\sqrt{n_{\mathcal{I}} m_{yy}^{\mathcal{I}} - (e_y^{\mathcal{I}})^2}}. \quad (10)$$

**Privacy protection.** Although accessing the one-dimensional statistics already makes it difficult for an adversary to reconstruct the original data, we aim to provide stronger privacy guarantees. Since the aggregation requires only summation, we employ Homomorphic Encryption (HE) (Paillier, 1999), which ensures that no individual client statistics (e.g., $e_x^{(k)}$, $m_{xy}^{(k)}$) are exposed during communication. Due to space limit, we refer the reader to Appendix B for related work and Appendix D for the complete procedure. This process fully preserves the privacy of client data.

**Client subset selection strategy.** The remaining problem is how to select a subset of clients that maximizes the power of the global test. Rather than relying on computationally expensive permutation procedures (Good, 2013), we directly use the summary statistic computed once by each client, which already captures the essential dependence information. Aggregating these summaries provides a practical approximation of the null distribution and yields a power score that guides subset selection. Importantly, our experiments confirm that this strategy is effective in practice (see Appendix I.3). Given such a power score, a natural idea is to evaluate different subsets of clients and choose the one with the highest score. One straightforward approach is to enumerate all nonempty subsets and evaluate their power scores, selecting the subset with the highest score. However, this approach quickly becomes computationally intractable as the number of clients $K$ grows, since the total number of subsets is $2^K$. To address this challenge, we propose a soft relaxation that converts the discrete optimization problem into a continuous one. Specifically, we assign each client $k \in [K]$

a learnable parameter $p_k \in [0, 1]$, define $n_{\mathcal{P}} = \sum_{k \in [K]} p_k n_k$, and rewrite the corresponding aggregated statistics as

$$e_x^{\mathcal{P}} = \sum_{k \in [K]} p_k e_x^{(k)}, \; m_{xy}^{\mathcal{P}} = \sum_{k \in [K]} p_k m_{xy}^{(k)}, \; \rho_{xy}^{\mathcal{P}} = \frac{n_{\mathcal{P}} m_{xy}^{\mathcal{P}} - e_x^{\mathcal{P}} e_y^{\mathcal{P}}}{\sqrt{n_{\mathcal{P}} m_{xx}^{\mathcal{P}} - (e_x^{\mathcal{P}})^2} \sqrt{n_{\mathcal{P}} m_{yy}^{\mathcal{P}} - (e_y^{\mathcal{P}})^2}}. \quad (11)$$

This continuous optimization problem can then be efficiently solved using gradient-based methods. This relaxation transforms the subset selection problem into a continuous optimization task, which can be efficiently solved with gradient-based methods. However, compared with the discrete case, the privacy-preserving component requires additional refinement: homomorphic encryption (HE) can still be applied to computing the gradients of the aggregated quantities with respect to $p_k$, thereby enabling each client to privately update its local weight parameter $p_k$ (see Appendix D for details).

### 4.4 THE OVERALL FRAMEWORK

Above, we introduced the intra-client modules and the aggregation process in detail. We now present the complete framework, named **FedIT-CS** (Federated Independence Testing with Copula Alignment and Stacking Aggregation). Depending on the aggregation strategy, we distinguish two main variants: **FedIT-CS-M**, which enables maximum power selection over client subsets, and **FedIT-CS-ML**, which further generalizes the procedure by allowing mixed linear aggregation, while achieving linear-time complexity with respect to the number of clients.

Table 1: Comparison of FUIT (Li et al., 2024) and the variants of our proposed FedIT-CS.

| Method | FUIT | FedIT-CS-S (Ours) | FedIT-CS-M (Ours) | FedIT-CS-ML (Ours) |
|---|---|---|---|---|
| Theoretical soundness | ✗ | ✗ | ✓ | ✓ |
| Privacy protection | ✗ | ✓ | ✓ | ✓ |
| Maximum power selection | ✗ | ✗ | ✓ | ✓ |
| Local computation cost | $\mathcal{O}(nh^2)$ | $\mathcal{O}(Bn\log n + Bnh^2)$ | $\mathcal{O}(Bn\log n + Bnh^2)$ | $\mathcal{O}(Bn\log n + Bnh^2)$ |
| Aggregation cost | $\mathcal{O}(Kh^2)$ | $\mathcal{O}(KB)$ | $\mathcal{O}(2^K B)$ | $\mathcal{O}(KB)$ |
| Communication cost | $\mathcal{O}(Kh^2)$ | $\mathcal{O}(KB)$ | $\mathcal{O}(KB)$ | $\mathcal{O}(KB)$ |

**Permutation-based testing.** To obtain $p$-value for hypothesis testing, we adopt a permutation-based procedure (Good, 2013). Formally, we generate $B$ permuted samples $(\sigma_t(\boldsymbol{x}), \boldsymbol{y})$ with $t \in [B]$, where each $\sigma_t$ is an independent derangement. These permuted pairs simulate samples under $\mathcal{H}_0$. Each client can perform this procedure independently, producing local null samples which, after intra-client transformation obtain copula output, are aggregated to compute the global test statistic. This enables an approximation of the null distribution of the aggregated test.

**Sample splitting.** A key detail is that we cannot use the same data both for learning the aggregation strategy and for performing the test, as this would invalidate Type I error control. To address this, we adopt a straightforward data-splitting strategy (Jitkrittum et al., 2017; Liu et al., 2020), which is simple and direct but inevitably reduces statistical power. Actually, more advanced strategies such as (Schrab et al., 2022; Kübler et al., 2020) could be considered, and we leave this as an important direction for further work.

For comparison with FUIT, we also introduce a naive variant, **FedIT-CS-S**, which aggregates copula outputs by direct pooling to compute covariance, equivalent to applying FedIT-CS-M over the entire client set without selection. This variant does not require sample splitting, but sacrifices theoretical guarantees. Table 1 summarizes all variants of our proposed framework. The table highlights that only FedIT-CS-M and FedIT-CS-ML achieve both *theoretical soundness* and *maximum power selection*, while all FedIT-CS variants ensure *privacy protection*. From a computational perspective, FedIT-CS-S and FedIT-CS-ML achieve efficient aggregation with cost $\mathcal{O}(KB)$, whereas FedIT-CS-M incurs exponential cost $\mathcal{O}(2^K B)$ due to subset enumeration.

**Algorithm.** The overall procedure is summarized in Alg. 1. As a preprocessing step, each client splits its data into training $\boldsymbol{Z}_k^{tr}$ and testing $\boldsymbol{Z}_k^{te}$ (Line 1). The test consists of two phases: (i) intra-client dependency modeling and client subset selection using the training data (Lines 2–5), and (ii) a permutation test with the learned aggregation weights to compute the $p$-value and decide inde-

pendence on the testing data (Lines 6–9). The computational complexity of different variants is summarized in Table 1 and specific privacy-preserving schemes can be applied.

---

**Algorithm 1** FedIT-CS Framework

---

**Input:** Number of clients $K$, federal samples $\boldsymbol{Z}_k = \{(x_i^{(k)}, y_i^{(k)})\}_{i=1}^{n_k}, k \in [K]$, the number of feature sampling $h$, significance level $\alpha$, the number of permutation $B$.
**Output:** Decision on $X \perp\!\!\!\perp Y$ or $X \not\!\perp\!\!\!\perp Y$.

1: Split client data into training and testing sets: $\boldsymbol{Z}_k = \boldsymbol{Z}_k^{tr} \cup \boldsymbol{Z}_k^{te}$.
2: ◁ **Intra-client dependency modeling and subset selection with $\boldsymbol{Z}_k^{tr}, k \in [K]$.**
3: Apply the first copula transform, random projections, and CCA as in Eq. (8).
4: Apply the second copula transform and aggregate with Eqs. (10) or (11).
5: Obtain the optimized aggregation weights $\boldsymbol{p}^* = (p_1^*, p_2^*, \ldots, p_K^*)$.
6: ◁ **Permutation test with $\boldsymbol{p}^*$ on $\boldsymbol{Z}_k^{te}, k \in [K]$.**
7: Generate $B$ permuted samples $(\sigma_t(\boldsymbol{x}), \boldsymbol{y})$, $t \in [B]$, then each perform intra-client transform.
8: Compute the aggregated statistic sequence $\{\rho_{xy}^{\mathcal{P}}, \rho_{xy}^{\mathcal{P},\sigma_1}, ..., \rho_{xy}^{\mathcal{P},\sigma_B}\}$.
9: Calculate the $p$-value by $p$-value$= [1 + \sum_{t=1}^B \mathbb{I}\{\rho_{xy}^{\mathcal{P},\sigma_t} \geq \rho_{xy}^{\mathcal{P}}\}]/[1 + B]$.
10: Reject $X \perp\!\!\!\perp Y$ if $p$-value $\leq \alpha$; otherwise accept $X \perp\!\!\!\perp Y$.

---

# 5 THEORETICAL ANALYSIS

Let the aggregation algorithm be denoted by $\mathcal{A}$, which determines the optimal weights $p_k$, $k \in [K]$. Note that when $p_k \in \{0, 1\}$, this corresponds to the Fed-CS-M strategy, while allowing $p_k \in [0, 1]$ corresponds to Fed-CS-ML. We now consider the case where $\mathcal{A}$ is theoretically optimal. In the asymptotic regime where the sample size and the number of random features are large enough, let the resulting aggregated coefficient be denoted by $\rho_{\mathcal{A}}$. Then, we have the following theorem to show that $\rho_{\mathcal{A}}$ is theoretically sound, with the proof given in Appendix F.

**Theorem 4** (Soundness of aggregated statistics). *Let $\rho_{\mathcal{A}}$ denote the aggregated coefficient obtained by the aggregation algorithm $\mathcal{A}$. Assume $\mathcal{A}$ is theoretically optimal. Then, under the null hypothesis $\mathcal{H}_0$, we have $\rho_{\mathcal{A}} = 0$, whereas under the alternative hypothesis $\mathcal{H}_1$, we have $\rho_{\mathcal{A}} > 0$.*

This result guarantees that our aggregated coefficient is theoretically correct in the idealized setting where sample estimation error and random approximation error are negligible and where the optimization procedure (in the case of FedIT-CS-ML) converges sufficiently well. In practice, by Theorems 2 and 3, the estimation and approximation errors vanish at rates $\mathcal{O}(1/\sqrt{n})$ and $\mathcal{O}(1/\sqrt{h})$ with high probability, thus acceptable. Moreover, the optimization step is effective due to the clear signal structure, as further validated in our experiments. Overall, this theoretical guarantee underpins the reliability of our framework, ensuring that both Type I and Type II error controls are meaningful. Next, we establish the Type I error bound of our test, with the proof being provided in Appendix G.

**Theorem 5** (Type I error bound). *Assume the null hypothesis $\mathcal{H}_0$ is true. For any significance level $\alpha \in (0, 1)$, the bound for the Type I error is given by*

$$\mathbb{P}(p\text{-value} \leq \alpha | \mathcal{H}_0) \leq \alpha. \tag{12}$$

This establishes the validity of our test. Together with the theoretical soundness of the aggregated statistic, this result provides a strong foundation for the reliability of our framework.

# 6 PERFORMANCE EVALUATION

We compare the following tests: FUIT (Li et al., 2024), FedIT-CS-S, FedIT-CS-M, and FedIT-CS-ML. To further evaluate the potential loss of statistical power due to sample splitting, we additionally include two variants, FedIT-CS-M-F and FedIT-CS-ML-F, where the aggregation strategy is trained on extra data that is not used for testing. For fairness, all methods are implemented with the number of random features fixed at $h = 10$, the significance level set to $\alpha = 0.05$, and the data split ratio set to 0.2. We evaluate the methods on both synthetic and real-world datasets. Due to space limit,

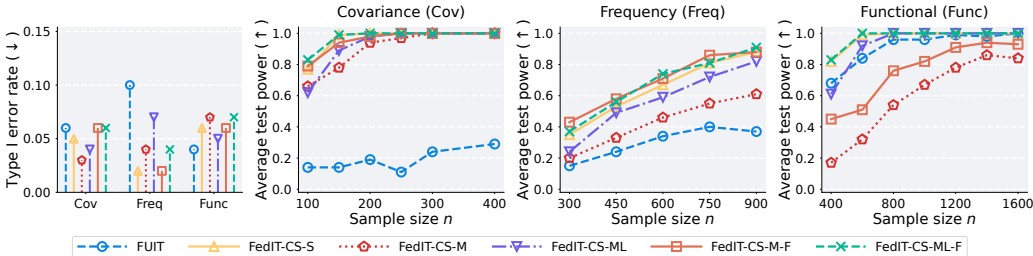

Figure 4: Left: The average Type I error rate on three synthetic datasets. The other three plots: The results of average test power on these three datasets.

detailed setup and additional results are provided in Appendices H and I, including experiments under more synthetic settings (with varying noise distributions and function generation processes), evaluations with larger numbers of clients, and comparisons of running time.

**Performance on synthetic datasets.** We consider three client heterogeneous scenarios. (i) *Covariance*: linear dependency with client-specific correlation coefficients 0.5, -0.5, and 0.02; sample size ratio $n_1 : n_2 : n_3 = 1 : 1 : 2$, with $n_1$ ranging from 100 to 400. (ii) *Frequency*: sinusoid model $(X, Y) \propto 1 + \sin(\omega x) \sin(\omega y)$ with $\omega = 2, 3, 4$ across clients; sample size ratio $n_1 : n_2 : n_3 = 1 : 1 : 1$, with $n_1$ from 300 to 900. (iii) *Functional*: nonlinear relations $Y = f(X) + \epsilon$, $f \in \{\sin(\cdot), \cos(\cdot), (\cdot)^2\}$ with $\epsilon \sim \mathcal{N}(0, 1)$; sample size ratio $n_1 : n_2 : n_3 = 4 : 2 : 1$, with $n_1$ from 400 to 1600. Type I error rate is evaluated using permuted samples. For each point, perform 100 repeated randomized experiments and report the average result. In all cases, figures are plotted with $n_1$.

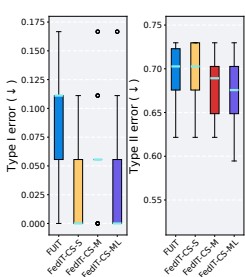

Figure 3: The results on Sachs.

Results are presented in Fig. 4. Except for FUIT on the Frequency setting, all methods successfully control the Type I error rate. On both the Covariance and Frequency settings, our approach consistently outperforms FUIT, which can be attributed to the effectiveness of our aggregation strategy. Comparing the variants with and without additional training data (-F vs. non -F), we observe that data splitting indeed reduces statistical power, though FedIT-CS-ML still achieves better performance than FUIT. Interestingly, despite being designed primarily for efficiency, our linear-time variant FedIT-CS-ML outperforms FedIT-CS-M. This improvement may stem from the greater flexibility of its solution space, which provides additional benefits to strength dependency signal.

**Performance on real dataset.** For real-world evaluation, we use the Sachs dataset (Sachs et al., 2005) under seven perturbation conditions, treating each condition as a distinct client. This dataset is widely used in independence testing (Zhang et al., 2023b) (see the Appendix H for visualization of the distributions). The network consists of 11 nodes, yielding 55 node pairs: 18 independent (for Type I error evaluation) and 37 dependent (for Type II error evaluation). In each run, we randomly select 3 out of the 7 clients, evaluate all pairs, and compute the average result. This procedure is repeated 50 times with new client selections, and the results are reported. Results are shown in Fig. 3. Compared with FUIT, all variants of our method achieve tighter control of Type I error while simultaneously reducing Type II error, thereby demonstrating stronger detection power. These results provide empirical evidence supporting the theoretical advantage of our framework.

## 7 CONCLUSION

This paper presents FedIT-CS (Federated Independence Testing with Copula Alignment and Stacking Aggregation), a framework that overcomes both theoretical and practical challenges of client heterogeneity in federated testing. By analyzing the limitations of existing methods, FedIT-CS introduces copula-based marginal alignment and stacking-based aggregation to ensure validity with enhanced power, while also providing efficient and privacy-preserving implementations. Extensive experiments confirm its superiority over prior approaches. An interesting direction for future work is to extend FedIT-CS to conditional independence testing, further broadening its applicability.

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

# Appendix Organization

## A  LIST OF SYMBOLS AND NOTATIONS

| | |
|---|---|
| $\mathcal{O}, o$ | big, small O notion |
| $\mathcal{O}_p, o_p$ | big, small O notion in probability |
| $i.i.d.$ | independent and identically distributed |
| $\mathbb{R}$ | the set of real numbers |
| $\mathcal{B}(\mathbb{R})$ | Borel $\sigma$-algebra on $\mathbb{R}$ |
| $\mathbb{P}_X$ | marginal distribution of $X$ |
| $\mathbb{P}_{XY}$ | joint distribution of $X, Y$ |
| $F_X$ | cumulative distribution function (cdf) of $X$ |
| $\mathbb{E}[X]$ | expectation of $X$ |
| $\text{Var}(X)$ | variance of $X$ |
| $\text{Cov}(X, Y)$ | covariance of $X, Y$ |
| $X \perp\!\!\!\perp Y$ | random variables $X, Y$ are independent |
| $X \not\perp\!\!\!\perp Y$ | random variables $X, Y$ are not independent |
| $\text{Tr}(\cdot)$ | the trace of a square matrix |
| $\mathbf{1}$ | an vector of all ones |
| $\mathbf{H}$ | centering matrix define as $\mathbf{H} = \mathbf{I} - \frac{1}{n}\mathbf{1}\mathbf{1}^T$ |
| $\odot$ | element-wise product |
| $[n]$ | denotes the set $\{1, 2, ..., n\}$ |
| $\rho$ | Pearson's correlation coefficient |
| $\Gamma(\cdot)$ | Gamma function |
| $\times$ | the product symbol of topological space |
| $\|\cdot\|_2$ | spectral norm |
| $\|\cdot\|_F$ | Frobenius norm |
| $\stackrel{d}{=}$ | equality in distribution |

## B  RELATED WORK

**Independence testing.** Traditional correlation-based measures such as Pearson's coefficient (Cohen et al., 2009) and Kendall's $\tau$ capture only linear or monotonic associations, and therefore fail to detect general nonlinear dependencies. To characterize more complex relationships, and handle more high-dimensional settings (Liu et al., 2022; Zhang et al., 2023c; Zhang & Zhu, 2023), a wide range of nonlinear dependence measures have been developed. These methods can be broadly categorized into three groups. (i) Rank-based methods. Chatterjee (2021) extends traditional correlation ideas by exploiting ranks, making it robust to outliers and invariant under monotone transformations of the data. (ii) Distance-based methods. Popular representatives include distance covariance and related measures (Székely et al., 2007; Lyons, 2013; Székely & Rizzo, 2013; Ren et al., 2023), which employ characteristic functions to quantify dependence. These methods are flexible and effective for general nonlinear settings. (iii) Kernel-based methods. This family derives dependence criteria

from cross-covariance operators in reproducing kernel Hilbert spaces (RKHS). Early examples include Kernel Canonical Correlation (KCC) (Bach & Jordan, 2002), which maximizes the correlation between feature maps, and Constrained Covariance (COCO) (Gretton et al., 2005), which removes normalization constraints. The most widely used kernel-based approach is the Hilbert–Schmidt Independence Criterion (HSIC) (Gretton et al., 2007), which measures dependence via the squared Hilbert–Schmidt norm of the cross-covariance operator. Follow-up work has further improved HSIC by accelerating computation for large-scale data (Zhang et al., 2018) and by optimizing kernel parameters to enhance test power (Jitkrittum et al., 2017; Ren et al., 2024; Xu et al., 2025).

Despite extensive progress in classical settings, independence testing under federated environments, especially under client heterogeneity remains underexplored. Motivated by this gap, we systematically investigate the problem of Federated Independence Testing (FedIT). We identify the key challenges that arise in this setting and propose a principled framework to address them.

**Federated causal discovery.** Unlike traditional causal discovery methods (Spirtes, 2001; Yu et al., 2019; Ng et al., 2024) that assume data are independent and identically distributed ($i.i.d.$), federated causal discovery (FCD) must contend with decentralized and often heterogeneous data (Zhou et al., 2022). Existing methods can be grouped into three categories: (i) score-based methods, which evaluate candidate graphs with predefined scoring functions (Huang et al., 2018; Ren et al., 2025) and search strategies (Tsamardinos et al., 2006; Chickering, 2003; 2020). For example, DARLIS (Ye et al., 2024) employs distributed simulated annealing, while PERI (Mian et al., 2023) builds on local GES (Chickering, 2003) with worst-case regret aggregation; (ii) continuous optimization-based methods, which reformulate structure learning as an optimization problem. NOTEARS (Zheng et al., 2018) pioneered this in the centralized case, and its federated extensions include NOTEARS-ADMM (Ng & Zhang, 2022) using the alternating direction method of multipliers, FedDAG (Gao et al., 2023) based on the FedAvg (McMahan et al., 2016) paradigm, and Fed-CDI (Abyaneh et al., 2022) incorporating intervention-aware aggregation; and (iii) constraint-based methods, which rely on conditional independence tests (Zhang et al., 2012; 2017; Pogodin et al., 2024) and algorithms such as Peter Clark (PC) (Spirtes et al., 2000). In the federated setting, FedPC (Huang et al., 2023) aggregates skeletons and orientations via voting under homogeneous data, while FedCDH (Li et al., 2024) introduces a federated conditional independence test (FCIT) and a federated independent change principle (FICP) to handle heterogeneity, achieving state-of-the-art results in FCD.

Although FCD has made progress in handling heterogeneous data, a systematic study of FedIT is still lacking, even though its reliability is critical for the effectiveness of FCD methods themselves. In particular, the federated independence test used in FedCDH, referred to as FUIT, adopts a naive feature space aggregation strategy. In this work, we revisit FUIT and show that such aggregation suffers from inherent theoretical flaws and practical power degradation under heterogeneity. To overcome these limitations, we propose a new framework that directly addresses these challenges.

**Privacy-preserving hypothesis testing.** Research on privacy-preserving statistical testing has received growing attention. (i) In the non-federated setting, many works have developed differentially private (DP) techniques (Dwork & Roth, 2014; Mironov, 2017). For example, Kazan et al. (2023) proposed a black-box framework to privatize arbitrary hypothesis tests; Priv-PC (Wang et al., 2020) designed DP algorithms for discrete data via sensitivity analysis of conditional Kendall's $\tau$ and Spearman's $\rho$, later extended to numerical data (Zhang et al., 2023a); Kusner et al. (2016) analyzed the sensitivity of HSIC; Kalemaj et al. (2024) added Laplace noise to regression residuals for conditional independence testing; and Kim & Schrab (2023) studied DP permutation tests for kernel-based methods, reducing noise while preserving power. In addition, homomorphic encryption has also been applied, e.g., Lauter et al. (2015) proposed a private $\chi^2$ test for independence testing. (ii) In the federated setting, Pang et al. (2023) developed a secure federated correlation test (FED-$\chi^2$) and entropy estimation by reformulating computations as frequency moment estimation and enabling aggregation through stable projections.

Overall, most existing approaches rely on differential privacy, while a few employ homomorphic encryption but remain limited to discrete settings such as the $\chi^2$ test. By contrast, our method can also apply to continuous and nonlinear scenarios. Through a client-side nonlinear transformation, we reduce aggregation to second-order moments in the FedIT task, which naturally aligns with homomorphic encryption and enables exact privacy-preserving computation.

## C  DETAILS OF FEDIT METHODS

In this section, we provide detailed explanations of the table presented in Sec. 4.4 of the main paper. For ease of reference, we reproduce the table below.

Table 2: Comparison of FUIT (Li et al., 2024) and our proposed FedIT-CS variants in terms of theoretical properties, privacy, power, and computational / communication costs.

| Method | FUIT | FedIT-CS-S (Ours) | FedIT-CS-M (Ours) | FedIT-CS-ML (Ours) |
|---|---|---|---|---|
| Theoretical soundness | ✗ | ✗ | ✓ | ✓ |
| Privacy protection | ✗ | ✓ | ✓ | ✓ |
| Maximum power selection | ✗ | ✗ | ✓ | ✓ |
| Local computation cost | $\mathcal{O}(nh^2)$ | $\mathcal{O}(Bn \log n + Bnh^2)$ | $\mathcal{O}(Bn \log n + Bnh^2)$ | $\mathcal{O}(Bn \log n + Bnh^2)$ |
| Aggregation cost | $\mathcal{O}(Kh^2)$ | $\mathcal{O}(KB)$ | $\mathcal{O}(2^K B)$ | $\mathcal{O}(KB)$ |
| Communication cost | $\mathcal{O}(Kh^2)$ | $\mathcal{O}(KB)$ | $\mathcal{O}(KB)$ | $\mathcal{O}(KB)$ |

The entries in Table 2 are derived as follows:

- **FUIT** (Li et al., 2024): This method directly aggregates local covariance matrices from all clients. As discussed in Sec. 3.2, such a naive aggregation lacks theoretical guarantees. Moreover, it provides neither privacy protection nor any form of client selection procedure. The local computation cost arises from computing the local covariance matrix $C_{xy}^{(k)} \in \mathbb{R}^{h \times h}$ based on features of $X$ and $Y$ with dimension $h \times n$. Since the aggregation step is simply a summation of $C_{xy}^{(k)}, k \in [K]$, the aggregation cost is $\mathcal{O}(Kh^2)$. Similarly, the communication cost is $\mathcal{O}(Kh^2)$, as each client must transmit its local covariance matrix $C_{xy}^{(k)}$ to the server.

- **FedIT-CS-S (Ours)**: This is the naive aggregation variant, which directly sums all local statistics (i.e., the second-order moments), equivalent to pooling copula outputs. Consequently, it also lacks theoretical guarantees and does not involve any selection procedure. However, since only aggregated moments are transmitted, the process preserves privacy (see Appendix D). The local computation cost includes copula transformation via marginal sorting $\mathcal{O}(n \log n)$, covariance computation $\mathcal{O}(nh^2)$, and CCA $\mathcal{O}(h^3)$, yielding a total complexity of $\mathcal{O}(n \log n + nh^2)$. Note that this procedure is repeated for each permutation sample. The aggregation cost is simply summing the client-level second moments, with complexity $\mathcal{O}(KB)$, and the communication cost is also $\mathcal{O}(KB)$ for transmitting these statistics to the server.

- **FedIT-CS-M (Ours)**: This variant extends FedIT-CS by enabling maximum power selection over client subsets. Specifically, it aggregates second-order moments from a chosen subset of clients and uses the resulting global statistic to optimize the selection strategy. Theoretical soundness is guaranteed by Theorem 4, and privacy protection holds as discussed in Appendix D. The local computation cost is the same as FedIT-CS-S, i.e., $\mathcal{O}(n \log n + nh^2)$, repeated for each permutation sample. In contrast, the aggregation step requires evaluating all possible client subsets, leading to $\mathcal{O}(2^K B)$ complexity, while the communication cost remains $\mathcal{O}(KB)$.

- **FedIT-CS-ML (Ours)**: This variant further generalizes the aggregation scheme by allowing mixed linear combinations of clients, rather than restricting to subset selection. The aggregation weights are optimized over a continuous domain, enabling more flexible and efficient power maximization. Theoretical soundness is established in Theorem 4, with privacy protection guaranteed as in Appendix D. The local computation cost remains the same as FedIT-CS-S, i.e., $\mathcal{O}(n \log n + nh^2)$ for each permutation sample. Unlike FedIT-CS-M, the aggregation complexity reduces to $\mathcal{O}(KB)$, since the continuous optimization is performed in linear time with respect to the number of clients. The communication cost also remains $\mathcal{O}(KB)$.

In addition, we also consider several auxiliary variants used in the experiments: **FedIT-CS-M-F**, **FedIT-CS-ML-F**, and **FedIT-CS-MB**. The "-F" versions differ from their counterparts by employing extra data for training the aggregation strategy, thereby quantifying the power loss of sample splitting in FedIT-CS-M and FedIT-CS-ML. For **FedIT-CS-MB**, which is not included in the main paper, our correlation-based modeling of power strategy is replaced by a permutation-based approach. As discussed in Sec. 4.3 and empirically validated in Appendix I.3, correlation coefficients already capture key dependency information efficiently. These variants are included mainly for auxiliary comparison in the experimental study, see Appendix I for the detailed results.

# D DETAILS OF THE HOMOMORPHIC ENCRYPTION PROCEDURE

In the main paper, we briefly introduced the use of Homomorphic Encryption (HE) to ensure privacy. Here, we provide a more complete description of this process. Recall that HE is a cryptographic method that enables computations to be performed directly on encrypted data without requiring decryption. The decrypted result of these operations is identical to performing the same calculations on the original plaintext. This property ensures privacy during processing, making HE particularly useful in scenarios where sensitive information must remain encrypted even while being utilized. Unlike DP, HE preserves privacy without sacrificing accuracy. Moreover, HE is especially suitable for our FedIT-CS framework, as each client only needs to share a few one-dimensional quantities; thus, the computational overhead introduced by HE remains relatively low.

We now describe the privacy-preserving component of our FedIT-CS framework. Since only linear correlation statistics need to be aggregated across clients, we can leverage additive homomorphic encryption, typically the Paillier cryptosystem (Paillier, 1999), to ensure that individual client contributions remain private. Formally, let $(pk, sk)$ denote a public–private key pair, and let $Enc(\cdot, pk)$ and $Dec(\cdot, sk)$ be the corresponding encryption and decryption functions. For two plaintext values $x$ and $y$, the additive homomorphic property guarantees that

$$x + y = Dec(Enc(x, pk) \star Enc(y, pk), sk), \tag{13}$$

where $\star$ denotes the addition operation in the encrypted space. Below, we illustrate how HE is applied in our framework. Prior to aggregation, each client $k \in [K]$ produces the copula samples $(r_x^{(k)}, r_y^{(k)})$ as the output of the within-client dependence modeling stage.

**Privacy protection in FedIT-CS-M**. We introduce the privacy protection procedure in FedIT-CS-M framework, which solves a discrete optimization problem to select a subset of clients. During aggregation, each client computes its local summary statistics from these samples and transmits only their encrypted versions to the server. Specifically, client $k$ computes:

$$e_x^{(k)} = \sum_{i=1}^{n_k} r_{x;i}^{(k)}, \ e_y^{(k)} = \sum_{i=1}^{n_k} r_{y;i}^{(k)}, \ m_{xy}^{(k)} = \sum_{i=1}^{n_k} r_{x;i}^{(k)} r_{y;i}^{(k)}, \ m_{xx}^{(k)} = \sum_{i=1}^{n_k} r_{x;i}^{(k)} r_{x;i}^{(k)}, \ m_{yy}^{(k)} = \sum_{i=1}^{n_k} r_{y;i}^{(k)} r_{y;i}^{(k)}. \tag{14}$$

and send the encrypted list below to the server

$$Enc(n_k), \ Enc(e_x^{(k)}), \ Enc(e_y^{(k)}), \ Enc(m_{xx}^{(k)}), \ Enc(m_{xy}^{(k)}), \ Enc(m_{yy}^{(k)}). \tag{15}$$

Then the server can calculate the encrypted aggregation results:

$$Enc(n_{\mathcal{I}}) = \sum_{k \in \mathcal{I}} Enc(n_k), \ Enc(e_x^{\mathcal{I}}) = \sum_{k \in \mathcal{I}} Enc(e_x^{(k)}), \ Enc(e_y^{\mathcal{I}}) = \sum_{k \in \mathcal{I}} Enc(e_y^{(k)}),$$
$$Enc(m_{xy}^{\mathcal{I}}) = \sum_{k \in \mathcal{I}} Enc(m_{xy}^{(k)}), \ Enc(m_{xx}^{\mathcal{I}}) = \sum_{k \in \mathcal{I}} Enc(m_{xx}^{(k)}), \ Enc(m_{yy}^{\mathcal{I}}) = \sum_{k \in \mathcal{I}} Enc(m_{yy}^{(k)}).$$

Here, all summations are performed directly in the encrypted space, not in the source domain. For simplicity, we stipulate that the precise semantics of these operations are automatically determined by the domain of the encrypted inputs, which avoids any ambiguity with their plaintext counterparts. The server then transfers the encrypted statistics to one randomly selected client for decryption:

$$n_{\mathcal{I}} = Dec(Enc(n_{\mathcal{I}})), \ e_x^{\mathcal{I}} = Dec(Enc(e_x^{\mathcal{I}})), \ e_x^{\mathcal{I}} = Dec(Enc(e_x^{\mathcal{I}})),$$
$$m_{xy}^{\mathcal{I}} = Dec(Enc(m_{xy}^{\mathcal{I}})), \ m_{xx}^{\mathcal{I}} = Dec(Enc(m_{xx}^{\mathcal{I}})), \ m_{yy}^{\mathcal{I}} = Dec(Enc(m_{yy}^{\mathcal{I}})). \tag{16}$$

Once the plaintext statistics are obtained, the client can calculate the result and send it to the server:

$$\rho_{xy}^{\mathcal{I}} = \frac{n_{\mathcal{I}} m_{xy}^{\mathcal{I}} - e_x^{\mathcal{I}} e_y^{\mathcal{I}}}{\sqrt{n_{\mathcal{I}} m_{xx}^{\mathcal{I}} - (e_x^{\mathcal{I}})^2} \sqrt{n_{\mathcal{I}} m_{yy}^{\mathcal{I}} - (e_y^{\mathcal{I}})^2}}. \tag{17}$$

Throughout the entire process, no client-specific statistics (e.g., $e_x^{(k)}$, $m_{xx}^{(k)}$ ) are exposed. The server only gets the final results, and the client selected for decryption knows the aggregated results but has no idea which local statistics are included in the aggregated results.

**Privacy protection in FedIT-CS-ML**. We introduce the privacy protection procedure in FedIT-CS-ML framework, which solves a continuous optimization problem using gradient-based methods. We first recall that some related aggregated statistics:

$$n_{\mathcal{P}} = \sum_{k \in [K]} p_k n_k, \;\; e_x^{\mathcal{P}} = \sum_{k \in [K]} p_k e_x^{(k)}, \;\; e_y^{\mathcal{P}} = \sum_{k \in [K]} p_k e_y^{(k)},$$
$$m_{xy}^{\mathcal{P}} = \sum_{k \in [K]} p_k m_{xy}^{(k)}, \;\; m_{xx}^{\mathcal{P}} = \sum_{k \in [K]} p_k m_{xx}^{(k)}, \;\; m_{yy}^{\mathcal{P}} = \sum_{k \in [K]} p_k m_{yy}^{(k)}. \tag{18}$$

Further, we denote

$$\mathfrak{N} := n_{\mathcal{P}} m_{xy}^{\mathcal{P}} - e_x^{\mathcal{P}} e_y^{\mathcal{P}}, \;\; \mathfrak{A} := n_{\mathcal{P}} m_{xx}^{\mathcal{P}} - (e_x^{\mathcal{P}})^2, \;\; \mathfrak{B} := n_{\mathcal{P}} m_{yy}^{\mathcal{P}} - (e_y^{\mathcal{P}})^2, \tag{19}$$

and thus the optimizing criterion is given by

$$\rho_{xy}^{\mathcal{P}} = \frac{n_{\mathcal{P}} m_{xy}^{\mathcal{P}} - e_x^{\mathcal{P}} e_y^{\mathcal{P}}}{\sqrt{n_{\mathcal{P}} m_{xx}^{\mathcal{P}} - (e_x^{\mathcal{P}})^2} \sqrt{n_{\mathcal{P}} m_{yy}^{\mathcal{P}} - (e_y^{\mathcal{P}})^2}} =: \frac{\mathfrak{N}}{\sqrt{\mathfrak{A}\mathfrak{B}}}. \tag{20}$$

Then the gradient of $\rho_{xy}^{\mathcal{P}} = \mathfrak{N}/\sqrt{\mathfrak{A}\mathfrak{B}}$ with respect to any $p_k, \; k \in [K]$ is given by

$$\frac{\partial \rho_{xy}^{\mathcal{P}}}{\partial p_k} = \frac{1}{\sqrt{\mathfrak{A}\mathfrak{B}}} \Big( d\mathfrak{N}_k - \frac{\mathfrak{N}}{2} \Big( \frac{d\mathfrak{A}_k}{\mathfrak{A}} + \frac{d\mathfrak{B}_k}{\mathfrak{B}} \Big) \Big), \tag{21}$$

where the terms

$$d\mathfrak{N}_k = \frac{\partial \mathfrak{N}}{\partial p_k} = n_k m_{xy}^{\mathcal{P}} + n_{\mathcal{P}} m_{xy}^{(k)} - e_x^{(k)} e_y^{\mathcal{P}} - e_x^{\mathcal{P}} e_y^{(k)},$$
$$d\mathfrak{A}_k = \frac{\partial \mathfrak{A}}{\partial p_k} = n_k m_{xx}^{\mathcal{P}} + n_{\mathcal{P}} m_{xx}^{(k)} - 2 e_x^{\mathcal{P}} e_x^{(k)}, \tag{22}$$
$$d\mathfrak{B}_k = \frac{\partial \mathfrak{B}}{\partial p_k} = n_k m_{yy}^{\mathcal{P}} + n_{\mathcal{P}} m_{yy}^{(k)} - 2 e_y^{\mathcal{P}} e_y^{(k)}.$$

Blow, we detail the optimization process under the protection of HE. In each communication round, client $k$ sends the encrypted local statistics to the server:

$$Enc(p_k n_k), \; Enc(p_k e_x^{(k)}), \; Enc(p_k e_y^{(k)}), \; Enc(p_k m_{xx}^{(k)}), \; Enc(p_k m_{xy}^{(k)}), \; Enc(p_k m_{yy}^{(k)}). \tag{23}$$

Then the server aggregates local statistics and sends them back to all clients:

$$Enc(n_{\mathcal{P}}) = \sum_{k \in [K]} Enc(p_k n_k), \qquad Enc(e_x^{\mathcal{P}}) = \sum_{k \in [K]} Enc(p_k e_x^{(k)}),$$
$$Enc(m_{xy}^{\mathcal{P}}) = \sum_{k \in [K]} Enc(p_k m_{xy}^{(k)}), \;\; Enc(e_y^{\mathcal{P}}) = \sum_{k \in [K]} Enc(p_k e_y^{(k)}), \tag{24}$$
$$Enc(m_{xx}^{\mathcal{P}}) = \sum_{k \in [K]} Enc(p_k m_{xx}^{(k)}), \;\; Enc(m_{yy}^{\mathcal{P}}) = \sum_{k \in [K]} Enc(p_k m_{yy}^{(k)}).$$

Once receives the aggregated results, client $k$ could calculate the decryption to obtain the following values:

$$n_{\mathcal{P}} = Dec(Enc(n_{\mathcal{P}})), \;\; e_x^{\mathcal{P}} = Dec(Enc(e_x^{\mathcal{P}})), \;\; e_x^{\mathcal{P}} = Dec(Enc(e_x^{\mathcal{P}})),$$
$$m_{xy}^{\mathcal{P}} = Dec(Enc(m_{xy}^{\mathcal{P}})), \;\; m_{xx}^{\mathcal{P}} = Dec(Enc(m_{xx}^{\mathcal{P}})), \;\; m_{yy}^{\mathcal{P}} = Dec(Enc(m_{yy}^{\mathcal{P}})). \tag{25}$$

Together with its local statistics, client $k$ could compute the gradient $\frac{\partial \rho_{xy}^{\mathcal{P}}}{\partial p_k}$ by replacing the corresponding values into Eq. (19) and Eq. (22). Throughout the entire process, no client-specific statistics (e.g., $e_x^{(k)}, m_{xx}^{(k)}$) are disclosed, which enables our gradient-based method to be computed without any privacy leakage.

# E PRELIMINARIES AND AUXILIARY LEMMAS

In this section, we provide a systematic overview of the entire procedure, introduce the notation used throughout, and establish several auxiliary lemmas. These intermediate results serve as building blocks for the proofs of our main theorems in the subsequent sections.

## E.1 ASSUMPTIONS

Below, we outline the assumptions required for our analysis. The first two assumptions are specific to the federated independence testing (FedIT) setting:

**Assumption 1** (Consistent dependence assumption). *We assume that the dependence relationship between $X_k$ and $Y_k$ is consistent across clients. That is, either all clients satisfy independence ($X_k \perp\!\!\!\perp Y_k$) or all exhibit dependence ($X_k \not\perp\!\!\!\perp Y_k$).*

**Assumption 2** (Heterogeneous clients assumption). *The dependence mechanism (e.g., strength or functional relationship) and the marginal distributions $\mathbb{P}_{X_k}$ and $\mathbb{P}_{Y_k}$ may vary across clients.*

Together, these assumptions define a realistic yet challenging regime: the global dependence status is common, but local distributions are heterogeneous. Our goal is to design a test that aggregates cross-client evidence to infer the shared dependence status while handling client heterogeneity.

Furthermore, for intra-client part, we impose a mild restriction on the class of functions of the Hirschfeld-Gebelein-Rényi (HGR) maximum correlation coefficient (Gebelein, 1941), defined as $\text{hgr}(X, Y) = \sup_{f,g} \rho(f(X), g(Y))$, where $\rho$ denotes Pearson's correlation.

**Assumption 3** (Function class assumption). *For each client $k \in [K]$, the optimal transformations $f_k$ and $g_k$ that maximize dependence, i.e. $\arg\max_{f_k,g_k} \rho(f_k(X_k), g_k(Y_k))$ are assumed to lie within a reproducing kernel Hilbert space (RKHS) $\mathcal{F}$ as in Theorem 3, which associated with a shift-invariant, positive semi-definite and bounded kernel $k(x, x') = \langle \phi(x), \phi(x') \rangle_{\mathcal{F}} \leq C$.*

This ensures that the transformations are sufficiently expressive to capture nonlinear dependencies, while also allowing tractable analysis and approximation through random Fourier features.

## E.2 COPULA PROPERTIES: MARGINAL UNIFORMITY AND CONVERGENCE

For completeness, we restate here some basic properties of copulas that will be used in the subsequent analysis. In particular, we recall the fact that copulas have uniformly distributed margins and summarize the convergence guarantees of their empirical estimators.

**Theorem 1** (Probability integral transform (Nelsen, 2006)). *For a random variable $X$ with cdf $F$, the random variable $U = F(X)$ is uniformly distributed on $[0, 1]$.*

The above result directly implies that the margins of any copula are uniformly distributed on $[0, 1]$, which forms the basis for the copula representation of dependence. Beyond this marginal property, it is also important to understand how well the empirical copula estimates converge to their population counterpart, since our method relies on finite-sample approximations.

**Theorem 2** (Convergence of the empirical copula Póczos et al. (2012)). *Let $\boldsymbol{F}$ be the copula transformation defined above, and $\boldsymbol{F}_n$ be the corresponding empirical copula transformation, then*

$$\mathbb{P}\left[\sup_{\boldsymbol{x} \in \mathbb{R}^d} \|\boldsymbol{F}(\boldsymbol{x}) - \boldsymbol{F}_n(\boldsymbol{x})\|_2 > \epsilon\right] \leq 2d \exp\left(-\frac{2n\epsilon^2}{d}\right). \tag{26}$$

Together, Theorems 1 and 2 establish that copulas not only provide uniform margins but also enjoy strong concentration guarantees for their empirical estimators. These properties will be repeatedly invoked in our subsequent theoretical analysis.

## E.3 RANDOM PROJECTION PROPERTIES: CONVERGENCE

We next recall a key result on the convergence behavior of random projections, which underpins their use in our framework. In particular, the following theorem shows that linear mixed models built on random features can approximate the optimal estimator within a controlled error that decays with both the sample size $n$ and the number of projections $h$ with high probability.

**Theorem 3** (Approximation with random projections Póczos et al. (2012))**.** *Let $p(w)$ be a distribution on $\Omega$ and $\sup_{x,w} |\phi(x; w)| \leq 1$. Let $\mathcal{F} = \{f(x) = \int_\Omega u(w)\phi(x; w)dw \mid |u(w)| \leq Cp(w)\}$. Draw $w_1, ..., w_h$ i.i.d from $p(w)$. Further let $\delta > 0$, and $c$ be some L-Lipschitz loss function, and consider data $\{x_i, o_i\}_{i=1}^n$ drawn i.i.d from some arbitrary $\mathbb{P}_{XO}$. The linear regression coefficient $u_1, ..., u_h$ for which $f_h(x) = \sum_{i=1}^h u_i\phi(x; w_i)$ minimizes the empirical risk $c(f_h(x), o)$ has a distance from the c-optimal estimator in $\mathcal{F}$ bounded by*

$$\mathbb{E}_\mathbb{P}[c(f_h(x), o)] - \min_{f \in \mathcal{F}} \mathbb{E}_\mathbb{P}[c(f(x), o)] \leq O\left(\left(\frac{1}{\sqrt{n}} + \frac{1}{\sqrt{h}}\right)LC\sqrt{\log\frac{1}{\delta}}\right) \tag{27}$$

*with probability at least $1 - 2\delta$.*

This result establishes that the approximation error decreases as the number of random features $h$ increases and as the sample size $n$ grows. Hence, random projections provide a computationally efficient way to approximate nonlinear function classes while retaining statistical guarantees.

### E.4 PROCEDURE AND PROPERTIES OF FEDIT-CS FRAMEWORK

In this section, we provide the preliminaries and several auxiliary lemmas that will facilitate the subsequent proofs. As a starting point, we restate the overall procedure and introduce the notation for the variables involved at each step. Specifically, for the $k$-th client, let $(X_k, Y_k)$ denote its input variables. The within-client computation for client $k \in [K]$ proceeds in four steps:

1. **First copula transform:** map $(X_k, Y_k)$ to the copulas $(Q_{X;k}, Q_{Y;k})$ using the marginal cdfs $F_X^{(k)}$ and $F_Y^{(k)}$. Note that $Q_{X;k}, Q_{Y;k} \sim \text{Uniform}[0, 1]$ by Theorem 1.
2. **Random projection:** transform $(Q_{X;k}, Q_{Y;k})$ into feature-space variables $(\Phi_{X;k}, \Phi_{Y;k})$ using random parameters $(w_X^{(k)}, b_X^{(k)})$ and $(w_Y^{(k)}, b_Y^{(k)})$.
3. **CCA step:** identify the canonical directions $u^{(k)}$ and $v^{(k)}$, and then project $(\Phi_{X;k}, \Phi_{Y;k})$ onto one-dimensional outputs $(\Psi_{X;k}, \Psi_{Y;k})$.
4. **Second copula transform:** align the margins of $(\Psi_{X;k}, \Psi_{Y;k})$ using their cdfs $(F_{\Psi_X}^{(k)}, F_{\Psi_Y}^{(k)})$, yielding the final copulas $(R_{X;k}, R_{Y;k})$. Also, $R_{X;k}, R_{Y;k} \sim \text{Uniform}[0, 1]$ by Theorem 1.

For simplicity, we first set aside the aggregation procedure and focus on analyzing the properties of the correlation coefficient $\rho(R_{X;k}, R_{Y;k})$ obtained within client $k$. As discussed in the main paper, this coefficient is closely related to the Hirschfeld–Gebelein–Rényi (HGR) Maximum Correlation Coefficient (Gebelein, 1941), which is defined as $\text{hgr}(X, Y) = \sup_{f,g} \rho(f(X), g(Y))$, where the supremum is taken over all Borel-measurable functions $f$ and $g$ with finite variance, and $\rho$ denotes Pearson's correlation. The main difference is that, in our case, the optimal transformations $f$ and $g$ are restricted to the function class associated with a Reproducing Kernel Hilbert Space (RKHS), namely $\mathcal{F} = \{f(x) = \int_\Omega u(w)\phi(x; w)dw \mid |u(w)| \leq Cp(w)\}$ as specified in Assumption 3.

A well-known result is that $\text{hgr}(X, Y)$ satisfies seven desirable properties (Rényi, 1959):

1. $\text{hgr}(X, Y)$ is defined for any pair of non-constant random variables $X$ and $Y$.
2. $\text{hgr}(X, Y) = \text{hgr}(Y, X)$.
3. $0 \leq \text{hgr}(X, Y) \leq 1$.
4. $\text{hgr}(X, Y) = 0$ iff $X$ and $Y$ are statistically independent.
5. For bijective Borel-measurable functions $f, g : \mathbb{R} \to \mathbb{R}$, $\text{hgr}(X, Y) = \text{hgr}(f(X), g(Y))$.
6. $\text{hgr}(X, Y) = 1$ if for Borel-measurable functions $f$ or $g$, $Y = f(X)$ or $X = g(Y)$.
7. If $(X, Y) \sim \mathcal{N}(\mu, \Sigma)$, then $\text{hgr}(X, Y) = |\rho(X, Y)|$, where $\rho$ is the correlation coefficient.

Since $\text{hgr}(X, Y)$ is invariant under bijective marginal transformations, the copula transform does not alter its value. Therefore, the coefficient $\rho(R_{X;k}, R_{Y;k})$ inherits these desirable properties. We summarize the key results in the following lemma.

**Lemma 1** (Intra-client dependency properties)**.** *Under Assumptions 1–3, for each client $k \in [K]$, the dependence coefficient $\rho(R_{X;k}, R_{Y;k})$ correctly characterizes dependence. Specifically, under $\mathcal{H}_0$, we have $\rho(R_{X;k}, R_{Y;k}) = 0$, whereas under $\mathcal{H}_1$, we have $\rho(R_{X;k}, R_{Y;k}) > 0$.*

*Proof.* This follows directly from the discussion above on the properties of hgr and the restricted function class in Assumption 3. □

Moreover, we can further characterize the output distribution under $\mathcal{H}_0$, as stated in the next lemma.

**Lemma 2** (Output copula under $\mathcal{H}_0$). *Under Assumptions 1–3, for each client $k \in [K]$, the transformed outputs satisfy $R_{X;k}, R_{Y;k} \sim Uniform([0,1] \times [0,1])$.*

*Proof.* By Lemma 1, under $\mathcal{H}_0$, $R_{X;k}$ and $R_{Y;k}$ are independent. In addition, the copula transform ensures that each marginal is uniformly distributed on $[0,1]$. Combining independence with marginal uniformity yields the stated result. □

Above, our discussion has focused on the idealized theoretical setting, where the estimation error and approximation error are ignored. We now turn to the practical case with finite samples and examine how the computation proceeds in this setting. For clarity, we restate the entire within-client procedure, along with the notation used at each step:

1. **First copula transform:** given the input sample $\{(x_i^{(k)}, y_i^{(k)})\}_{i=1}^{n_k}$ corresponding to $(X_k, Y_k)$, compute the copula $\{(q_{x;i}^{(k)}, q_{y;i}^{(k)})\}_{i=1}^{n_k}$ using the empirical marginal cdfs $F_{x;n_k}^{(k)}$ and $F_{y;n_k}^{(k)}$.

2. **Random projection:** map $\{(q_{x;i}^{(k)}, q_{y;i}^{(k)})\}_{i=1}^{n_k}$ into features $\{(\phi_{x;i}^{(k)}, \phi_{y;i}^{(k)})\}_{i=1}^{n_k}$ using sampling parameters $(w_{x;h}^{(k)}, b_{x;h}^{(k)})$ and $(w_{y;h}^{(k)}, b_{y;h}^{(k)})$, where $h$ denotes the number of random projections.

3. **CCA step:** compute the canonical directions $u_h^{(k)}$ and $v_h^{(k)}$, and project $\{\phi_{x;i}^{(k)}, \phi_{y;i}^{(k)}\}_{i=1}^{n_k}$ to obtain one-dimensional outputs $\{\psi_{x;i}^{(k)}, \psi_{y;i}^{(k)}\}_{i=1}^{n_k}$.

4. **Second copula transform:** apply the empirical cdfs $F_{\psi_x;n_k}^{(k)}$ and $F_{\psi_y;n_k}^{(k)}$ to $\{(\psi_{x;i}^{(k)}, \psi_{y;i}^{(k)})\}_{i=1}^{n_k}$, yielding the final copula samples $\{(r_{x;i}^{(k)}, r_{y;i}^{(k)})\}_{i=1}^{n_k}$, which are approximately uniform on $[0,1]$.

The above procedure can be applied to the permuation case as well. Following the notation in the main paper, let $\sigma_t$ denote the $t$-th permutation with $t \in [B]$. For each client $k$, define the sample vectors

$$\boldsymbol{x}^{(k)} = (x_1^{(k)}, x_2^{(k)}, \ldots, x_{n_k}^{(k)}), \quad \boldsymbol{y}^{(k)} = (y_1^{(k)}, y_2^{(k)}, \ldots, y_{n_k}^{(k)}).$$

Then $(\sigma_t \boldsymbol{x}^{(k)}, \boldsymbol{y}^{(k)})$ denotes the permuted sample pair. Since permutation only changes the order of observations but not their empirical distribution, we have the following result.

**Lemma 3** (Cumulative distribution function under permutation). *For all $\sigma_t$, $t \in [B]$, the permuted samples yield the same empirical cumulative distribution function $F_{x;n_k}^{(k)}$.*

*Proof.* By definition, the empirical cumulative distribution function (cdf) depends only on the multiset of sample values, not on their order. Since permutation reorders $\boldsymbol{x}^{(k)}$ without altering its elements, the resulting empirical cdf remains identical. □

This property ensures that permutations do not affect the marginal distributions. In addition, we can further exploit the exchangeability property of the permuted sequences.

**Lemma 4** (Exchangeability). *Let $\overset{d}{=}$ denote equality in distribution. Under $\mathcal{H}_0$, the sequence for client $k$, $(\boldsymbol{x}^{(k)}, \boldsymbol{y}^{(k)})$, $(\sigma_1 \boldsymbol{x}^{(k)}, \boldsymbol{y}^{(k)}), \ldots, (\sigma_B \boldsymbol{x}^{(k)}, \boldsymbol{y}^{(k)})$, is exchangeable.*

*Proof.* Under $\mathcal{H}_0$, $X_k$ and $Y_k$ are independent. Thus, for every permutation $\sigma_t$ with $t \in [B]$, the pair $(\sigma_t \boldsymbol{x}^{(k)}, \boldsymbol{y}^{(k)})$ has the same joint distribution as the original $(\boldsymbol{x}^{(k)}, \boldsymbol{y}^{(k)})$, i.e.,

$$(\sigma_t \boldsymbol{x}^{(k)}, \boldsymbol{y}^{(k)}) \overset{d}{=} (\boldsymbol{x}^{(k)}, \boldsymbol{y}^{(k)}).$$

Since each element in the sequence has the same distribution and the joint law is invariant under permutations of indices $t$, the sequence is exchangeable by definition. □

Together, Lemma 3 and Lemma 4 yield the following.

**Lemma 5** (Exchangeability of client-level output copulas). *Let $\overset{d}{=}$ denote equality in distribution. Fix a client $k$ and consider the mapping $\mathcal{T}$ defined by the four-step pipeline: (i) empirical copula transform, (ii) random projection, (iii) CCA step, and (iv) a second empirical copula transform. Under $\mathcal{H}_0$ (i.e., $X_k \perp\!\!\!\perp Y_k$), the sequence*

$$\mathcal{T}\big(\boldsymbol{x}^{(k)}, \boldsymbol{y}^{(k)}\big), \ \mathcal{T}\big(\sigma_1 \boldsymbol{x}^{(k)}, \boldsymbol{y}^{(k)}\big), \ldots, \mathcal{T}\big(\sigma_B \boldsymbol{x}^{(k)}, \boldsymbol{y}^{(k)}\big)$$

*is exchangeable.*

*Proof.* We verify that each step of $\mathcal{T}$ results in an exchangeable sequence under $\mathcal{H}_0$. As a begin, Lemma 4 show that the input sequence is exchangeable.

*Step (i): First copula transform.* By Lemma 3, the empirical cdfs $F_{x;n_k}^{(k)}$ and $F_{y;n_k}^{(k)}$ are invariant under permutations, so the copula samples are exchangeable.

*Step (ii): Random projection.* The projection parameters $(w_{\cdot;h}^{(k)}, b_{\cdot;h}^{(k)})$ are drawn i.i.d. from the same distribution. Applying them elementwise preserves exchangeability of the projected features.

*Step (iii): CCA step.* The canonical directions depend only on the covariance structure of the projected features, which is unaffected since the input feature distributions are the same. Thus, the resulting one-dimensional projections are also exchangeable.

*Step (iv): Second copula transform.* For the same sample input, the empirical cdfs are the same. As the outputs after CCA have the same distributions, thus in this step, exchangeable inputs yields exchangeable copula outputs.

Combining all steps, the sequence of outputs across permutations is exchangeable. □

As a consequence, the client-level output copula sequence is exchangeable, a property that we later exploit to establish test validity, in particular Type I error control.

# F    PROOF OF THEOREM 4

Next, we turn to the theoretical properties of the aggregated statistic. For clarity, the client-level results and the associated notation have been introduced in Appendix E. Here, we shift our focus to the aggregation step. Let the aggregation algorithm be denoted by $\mathcal{A}$, which determines the optimal weights $p_k$, $k \in [K]$. Note that when $p_k \in \{0, 1\}$, this corresponds to the Fed-CS-M strategy, while allowing $p_k \in [0, 1]$ corresponds to Fed-CS-ML. We now consider the case where $\mathcal{A}$ is theoretically optimal. In the asymptotic regime where the sample size and the number of random features are large enough, let the resulting aggregated coefficient be denoted by $\rho_{\mathcal{A}}$. Then, we have the following theorem to show $\rho_{\mathcal{A}}$ is theoretical sound.

**Theorem 4** (Soundness of aggregated statistic). *Let $\rho_{\mathcal{A}}$ denote the aggregated coefficient obtained by the aggregation algorithm $\mathcal{A}$. Assume $\mathcal{A}$ is theoretically optimal. Then, under the null hypothesis $\mathcal{H}_0$, we have $\rho_{\mathcal{A}} = 0$, whereas under the alternative hypothesis $\mathcal{H}_1$, we have $\rho_{\mathcal{A}} > 0$.*

*Proof.* We first consider the case under $\mathcal{H}_0$. According to Lemma 1 and 2, for each client $k \in [K]$, the dependence strength satisfies $\rho(R_{X;k}, R_{Y;k}) = 0$, and the output copulas follow the distribution $R_{X;k}, R_{Y;k} \sim \text{Uniform}([0,1] \times [0,1])$. Consequently, the second-order moments are identical across all clients. Therefore, regardless of how the aggregation algorithm assigns the weights, the aggregated coefficient remains zero, thus $\rho_{\mathcal{A}} = 0$. Next, we consider the case under $\mathcal{H}_1$. In this setting, by Lemma 1, each client exhibits a strictly positive dependence coefficient $\rho(R_{X;k}, R_{Y;k}) > 0$, reflecting the underlying dependence between $X_k$ and $Y_k$. Since the aggregation algorithm $\mathcal{A}$ is assumed to be theoretically optimal, it assigns weights $\{p_k\}_{k=1}^{K}$ in a way that maximizes the global statistic, thus $\rho_{\mathcal{A}} \geq \max_k \rho(R_{X;k}, R_{Y;k}) > 0$, which completes the whole proof. □

# G    PROOF OF THEOREM 5

In this section, we provide the proof for the Type I error bound of our proposed test. For clarity, the client-level exchangeability results and the associated notation have already been introduced in

Appendix E. Let the optimized aggregation weights be denoted by $\boldsymbol{p}^* = (p_1^*, p_2^*, \ldots, p_K^*)$. During the testing procedure, each client $k$ produces the empirical copula vectors $\boldsymbol{r}_x^{(k)}$ and $\boldsymbol{r}_y^{(k)}$ from its local data $(\boldsymbol{x}^{(k)}, \boldsymbol{y}^{(k)})$. For the corresponding permutation samples $(\sigma_t \boldsymbol{x}^{(k)}, \boldsymbol{y}^{(k)})$, we denote the resulting copula vectors as $\sigma_t \boldsymbol{r}_{\boldsymbol{x}}^{(k)}$ and $\sigma_t \boldsymbol{r}_{\boldsymbol{y}}^{(k)}$. Based on these definitions, we now establish the exchangeability property of the aggregated statistic.

**Proposition 1** (Exchangeablility of aggregated statistic)**.** *Let the optimized aggregation weights be* $\boldsymbol{p}^* = (p_1^*, \ldots, p_K^*)$, *fixed with respect to the permutations. For each client $k$, let* $(\boldsymbol{r}_x^{(k)}, \boldsymbol{r}_y^{(k)})$ *be the empirical copula vectors computed from* $(\boldsymbol{x}^{(k)}, \boldsymbol{y}^{(k)})$, *and let* $(\sigma_t \boldsymbol{r}_x^{(k)}, \sigma_t \boldsymbol{r}_y^{(k)})$ *denote the copula vectors obtained from the permuted pair* $(\sigma_t \boldsymbol{x}^{(k)}, \sigma_t \boldsymbol{y}^{(k)})$, *for $t \in [B]$. Also, the aggregated first/second moments and effective sample size for* $(\boldsymbol{x}^{(k)}, \boldsymbol{y}^{(k)}), k \in [K]$ *is given by*

$$
e_x^{\mathcal{P}} = \sum_{k=1}^{K} p_k^* e_x^{(k)}, \quad e_y^{\mathcal{P}} = \sum_{k=1}^{K} p_k^* e_y^{(k)}, \quad m_{xy}^{\mathcal{P}} = \sum_{k=1}^{K} p_k^* m_{xy}^{(k)},
$$

$$
m_{xx}^{\mathcal{P}} = \sum_{k=1}^{K} p_k^* m_{xx}^{(k)}, \quad m_{yy}^{\mathcal{P}} = \sum_{k=1}^{K} p_k^* m_{yy}^{(k)}, \quad n_{\mathcal{P}} = \sum_{k=1}^{K} p_k^* n_k, \tag{28}
$$

*and the aggregated Pearson-type statistic is then calculated as Eq. (11). For simplify, we denote* $\rho_{xy}^{\mathcal{P}, \sigma_0} := \rho_{xy}^{\mathcal{P}}$. *Under $\mathcal{H}_0$, the sequence* $\rho_{xy}^{\mathcal{P}, \sigma_0}$, $\rho_{xy}^{\mathcal{P}, \sigma_1}$, $\ldots$, $\rho_{xy}^{\mathcal{P}, \sigma_B}$, *constructed respectively from* $\{(\boldsymbol{r}_x^{(k)}, \boldsymbol{r}_y^{(k)})\}_{k=1}^{K}$ *and* $\{(\sigma_t \boldsymbol{r}_x^{(k)}, \sigma_t \boldsymbol{r}_y^{(k)})\}_{k=1}^{K}, t \in [B]$, *is exchangeable.*

*Proof. Step 1 (client-level exchangeability of copulas).* By Lemma 3 and Lemma 4, for each client $k$, the sequence $(\boldsymbol{r}_x^{(k)}, \boldsymbol{r}_y^{(k)})$, $\{(\sigma_t \boldsymbol{r}_x^{(k)}, \boldsymbol{r}_y^{(k)})\}_{t=1}^{B}$ is exchangeable under $\mathcal{H}_0$.

*Step 2 (exchangeability of moments).* The mappings

$$
(\boldsymbol{u}, \boldsymbol{v}) \mapsto e_u = \sum_i u_i, \quad m_{uu} = \sum_i u_i^2, \quad m_{uv} = \sum_i u_i v_i
$$

are sum operations over the sample index. Hence, for each $k$, the sequences $\{e_x^{(k,t)}\}_t$, $\{e_y^{(k,t)}\}_t$, $\{m_{xx}^{(k,t)}\}_t$, $\{m_{yy}^{(k,t)}\}_t$, and $\{m_{xy}^{(k,t)}\}_t$ computed from $(\sigma_t \boldsymbol{r}_x^{(k)}, \sigma_t \boldsymbol{r}_y^{(k)})$ are exchangeable.

*Step 3 (fixed-weight aggregation preserves exchangeability).* Because $e_\cdot^{\mathcal{P}}$, $m_{\cdot\cdot}^{\mathcal{P}}$, and $n_{\mathcal{P}}$ are fixed linear combinations of the client-level moments with deterministic weights $\{p_k^*\}$ that do not depend on $t$, the aggregated moment sequences across $t$ remain exchangeable.

*Step 4 (continuous mapping).* The map $(e_x^{\mathcal{P}}, e_y^{\mathcal{P}}, m_{xx}^{\mathcal{P}}, m_{yy}^{\mathcal{P}}, m_{xy}^{\mathcal{P}}, n_{\mathcal{P}}) \mapsto \rho_{xy}^{\mathcal{P}}$ is a measurable deterministic function. Therefore, by the continuous mapping principle for exchangeable arrays, the sequence $\{\rho_{xy}^{\mathcal{P}, \sigma_t}\}_{t=0}^{B}$ is exchangeable. $\qquad\square$

As a direct consequence, the Type I error of our proposed test is provably controlled.

**Theorem 5** (Type I error bound)**.** *Assume the null hypothesis $\mathcal{H}_0$ is true. For any significance level $\alpha \in (0, 1)$, the bound for the Type I error is given by*

$$
\mathbb{P}(\text{p-value} \leq \alpha | \mathcal{H}_0) \leq \alpha. \tag{29}
$$

*Proof.* For simplify, we also write $\mathbb{P}(\cdot | \mathcal{H}_0)$ as $\mathbb{P}_{\mathcal{H}_0}$. For any given $\alpha \in (0, 1)$, we have

$$
\mathbb{P}_{\mathcal{H}_0}(\text{p-value} \leq \alpha) = \mathbb{P}_{\mathcal{H}_0}\left( \frac{1 + \sum_{t=1}^{B} \mathbb{I}\{\rho_{xy}^{\mathcal{P}, \sigma_t} \geq \rho_{xy}^{\mathcal{P}, \sigma_0}\}}{1 + B} \leq \alpha \right)
$$

$$
\leq \mathbb{P}_{\mathcal{H}_0}\left( \sum_{t=1}^{B} \mathbb{I}\{\rho_{xy}^{\mathcal{P}, \sigma_t} \geq \rho_{xy}^{\mathcal{P}, \sigma_0}\} \leq \lfloor \alpha(1 + B) \rfloor \right). \tag{30}
$$

Since the sequence $\{\rho_{xy}^{\mathcal{P}, \sigma_t}\}_{t=0}^{B}$ is exchangeable, by the property of order statistics, we have

$$
\mathbb{P}_{\mathcal{H}_0}\left( \sum_{t=1}^{B} \mathbb{I}\{\rho_{xy}^{\mathcal{P}, \sigma_t} \geq \rho_{xy}^{\mathcal{P}, \sigma_0}\} \leq \lfloor \alpha(1 + B) \rfloor \right) = \frac{\lfloor \alpha(1 + B) \rfloor}{1 + B} \leq \alpha, \tag{31}
$$

which completes the proof. $\qquad\square$

## H   DETAILS OF EXPERIMENTAL SETUP AND ANALYSIS OF RESULTS

In this section, we provide detailed descriptions of the experimental settings used in the main paper, including dataset specifications and implementation details. We also present extended results and further analyses to complement the findings reported in the main paper.

**Implementation details.** All methods are implemented with the number of random features fixed at $h = 10$ and the significance level set to $\alpha = 0.05$. For FUIT, we use the median bandwidth setting. For methods requiring data splitting, namely FedIT-CS-M and FedIT-CS-ML, the split ratio is fixed at 0.2. For the two variants FedIT-CS-M-F and FedIT-CS-ML-F, where the aggregation strategy is trained on additional data not used for testing, the extra data are set to have the same size as the testing data by default. For FedIT-CS-ML and FedIT-CS-ML-F, which involve gradient-based optimization, the number of iterations is set to 100. And the permutation number $B$ is set to 100 by default. All experiments are conducted on the same hardware platform equipped with 8-core CPU.

### H.1   DETAILS ABOUT SYNTHETIC DATA EXPERIMENTS

Below, we provide the setup details of the synthetic datasets, which include three heterogeneous scenarios: covariance heterogeneity, frequency heterogeneity, and functional heterogeneity. These settings are designed to evaluate the capability of our method under different types of distributional shifts. For this part, the number of clients is $K = 3$.

***Covariance.*** We consider a heterogeneous scenario where clients follow distinct covariance structures. The data generation process is specified as follows:

- **Client 1:** $X \sim \mathcal{N}(0, 1), \quad Y = 0.5X + \epsilon_1, \ \epsilon_1 \sim \mathcal{N}(0, 1);$
- **Client 2:** $X \sim \mathcal{N}(0, 1), \quad Y = -0.5X + \epsilon_2, \ \epsilon_2 \sim \mathcal{N}(0, 1);$
- **Client 3:** $X \sim \mathcal{N}(0, 1), \quad Y = 0.02X + \epsilon_3, \ \epsilon_3 \sim \mathcal{N}(0, 1).$

The sample sizes follow the ratio $n_1 : n_2 : n_3 = 1 : 1 : 2$. We vary the sample size of Client 1 with $n_1 \in \{100, 150, 200, 250, 300, 400\}$, and scale the other clients proportionally.

***Frequency.*** We next consider a heterogeneous scenario where clients exhibit distinct frequency parameters. Specifically, we adopt the sinusoid model (Sejdinovic et al., 2013) with density

$$(X, Y) \sim \mathbb{P}_{xy}(x, y) \propto 1 + \sin(\omega x)\sin(\omega y), \quad (x, y) \in [-\pi, \pi] \times [-\pi, \pi], \tag{32}$$

where $\omega$ denotes the frequency. Larger $\omega$ values make the distribution closer to Uniform($[-\pi, \pi]^2$), thereby increasing the difficulty of detecting dependence for small sample sizes. We assign client-specific frequencies $\omega_1 = 2$, $\omega_2 = 3$, and $\omega_3 = 4$. The sample sizes follow the ratio $n_1 : n_2 : n_3 = 1 : 1 : 1$. We vary the sample size of Client 1 with $n_1 \in \{100, 150, 200, 250, 300, 400\}$, and scale the other clients proportionally.

***Functional.*** Finally, we consider a heterogeneous scenario where clients follow distinct functional relationships. The data generation procedure is defined as follows:

- **Client 1:** $X \sim \text{Uniform}(0, 1), \quad Y = \sin(X) + \epsilon_1, \ \epsilon_1 \sim \mathcal{N}(0, 1);$
- **Client 2:** $X \sim \text{Uniform}(0, 1), \quad Y = \cos(X) + \epsilon_2, \ \epsilon_2 \sim \mathcal{N}(0, 1);$
- **Client 3:** $X \sim \text{Uniform}(0, 1), \quad Y = X^2 + \epsilon_3, \ \epsilon_3 \sim \mathcal{N}(0, 1).$

The sample sizes for the three clients follow the ratio $n_1 : n_2 : n_3 = 4 : 2 : 1$. We vary the sample size of Client 1 with $n_1 \in \{400, 600, 800, 1000, 1200, 1400, 1600\}$, and scale the other clients proportionally. Note that in this case, all clients have strong intra-client dependency.

### H.2   DETAILS ABOUT REAL DATA EXPERIMENTS

**Sachs dataset.** To evaluate the effectiveness of our proposed method in real-world, we employed the well-known Sachs (Sachs et al., 2005) dataset under seven perturbation conditions: (i) anti-CD3 + anti-CD28, (ii) anti-CD3/CD28 + ICAM-2, (iii) anti-CD3/CD28 + U0126, (iv) anti-CD3/CD28 + AKT inhibitor, (v) anti-CD3/CD28 + G06976, (vi) anti-CD3/CD28 + Psitectorigenin, and (vii) anti-CD3/CD28 + LY294002. In our setting, each perturbation condition in the Sachs dataset is regarded as a distinct client. These perturbations cover both general T-cell activation and specific

pathway-targeted interventions, thereby enabling a diverse range of causal effects within the signaling network. The detailed biological functions of individual reagents are summarized in Table 3. The causal network is presented in Fig. 5. This network comprises 11 nodes and 18 arcs, and is commonly recognized as a benchmark ground truth. It has been extensively adopted in prior studies on causal discovery (Zhang et al., 2023b). The 11 nodes of this network form 55 distinct node pairs in total. Among these pairs, 18 are independent of each other, whereas the remaining 37 exhibit a dependent relationship. As an illustrative example, the nodes Plcg and PKC are independent, while the relationship between Plcg and itself is dependent. We further visualize the distributions of all 11 variables under each experimental condition, and the results are presented in Fig. 6, where each row corresponds to one perturbation condition and each column corresponds to one variable. *From the marginal distributions, we can observe that the variable distributions change under different perturbation conditions, which aligns with our heterogeneity assumption.*

Table 3: Summary of reagents employed in the perturbation conditions and their biological effects.

| Reagent | Class | Biological effect |
|---|---|---|
| Anti-CD3 | General | Activates T-cell receptor (TCR) signaling, initiating proximal signaling cascades. |
| Anti-CD28 | General | Provides co-stimulatory signal for T-cell activation, enhancing proliferation and cytokine production. |
| ICAM-2 | General | Triggers LFA-1 adhesion signaling and cooperates with CD3/CD28 to enhance AP-1 and NFAT activation. |
| U0126 | Specific | Noncompetitive inhibitor of MEK1/2; blocks Erk activation and arrests T-cell proliferation. |
| AKT inhibitor | Specific | Blocks AKT membrane translocation and phosphorylation, suppressing AKT-mediated survival signaling. |
| G06976 | Specific | Inhibits PKC isozymes; blocks PKC-mediated T-cell activation. |
| Psitectorigenin | Specific | Inhibits phosphoinositide hydrolysis and phosphoinositol turnover. |
| LY294002 | Specific | PI3K inhibitor; prevents subsequent activation of AKT. |

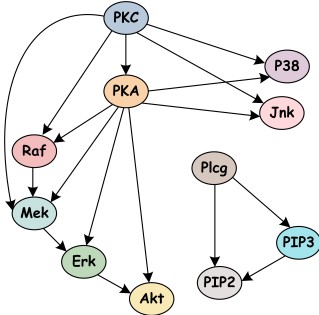

Figure 5: The causal graph of Sachs network.

**Experimental setup.** In our setting, each perturbation condition in the Sachs dataset is treated as a distinct client, yielding a total of seven clients. We compare our proposed methods against FUIT, FedIT-CS-S, FedIT-CS-M, and FedIT-CS-ML. At each iteration, 3 clients are randomly sampled from 7 clients, and we evaluate all 55 node pairs: 18 independent pairs for Type I error assessment and 37 dependent pairs for Type II error assessment. To ensure statistical reliability, this procedure is repeated 50 times with independent client selections, and we report the averaged performance across all trials. The results are summarized in Table 4.

**Performance and analysis.** Compared with FUIT, all variants of our method—FedIT-CS-S, FedIT-CS-M, and FedIT-CS-ML—achieve tighter control of Type I error while simultaneously reducing Type II error, thereby demonstrating stronger detection power. Notably, although the data splitting used in FedIT-CS-ML may degrade statistical power, it still achieves the best overall performance,

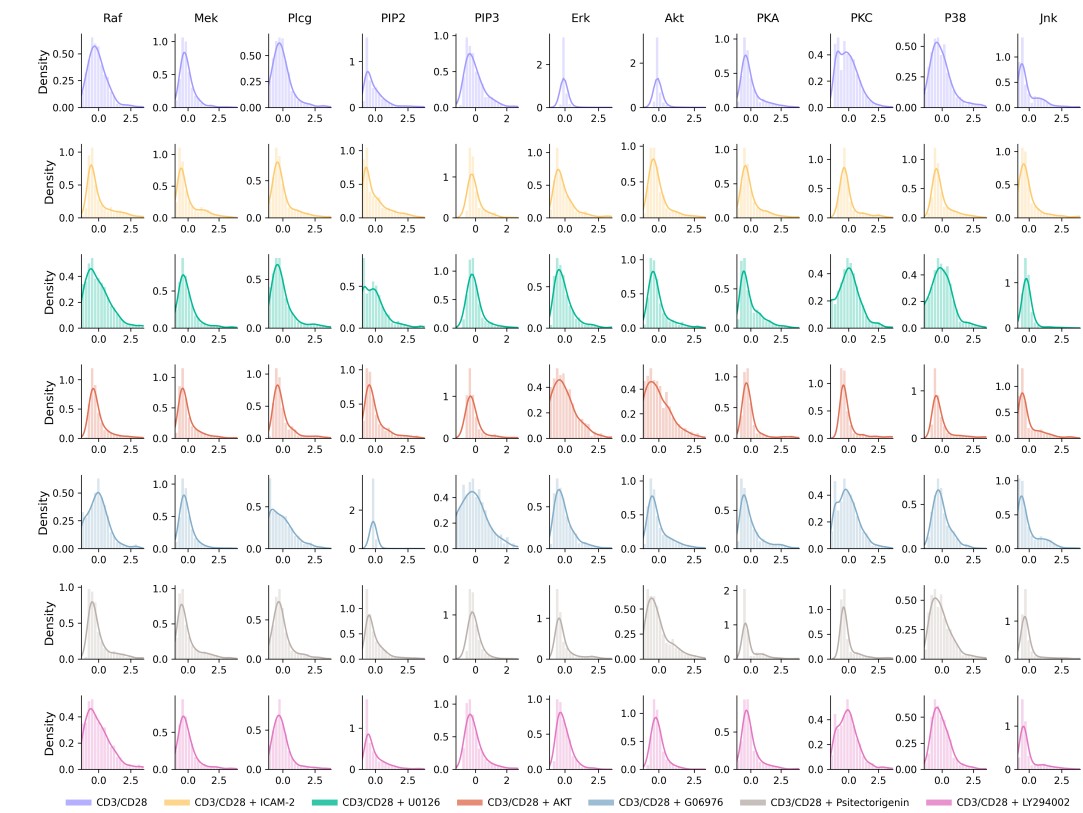

Figure 6: Distribution of the 11 variables across seven perturbation conditions in the Sachs dataset.

Table 4: Performance Comparison on the Sachs Dataset. Best in gray.

| Method | Type I error ($\downarrow$) | Type II error ($\downarrow$) |
|---|---|---|
| **FUIT** | $0.1011 \pm 0.0595$ | $0.6973 \pm 0.0276$ |
| **FedIT-CS-S (Ours)** | $0.0189 \pm 0.0307$ | $0.6968 \pm 0.0312$ |
| **FedIT-CS-M (Ours)** | $0.0589 \pm 0.0375$ | $0.6843 \pm 0.0298$ |
| **FedIT-CS-ML (Ours)** | $0.0333 \pm 0.0458$ | $0.6692 \pm 0.0396$ |

benefiting from our stacking-based aggregation strategy. These findings provide empirical evidence that supports the theoretical advantages of our framework.

# I    ADDITIONAL EXPERIMENT RESULTS

In this section, we present additional experimental results under a broader range of settings, including diverse distributional settings, varied functional relationships, and different numbers of clients. We also provide supplementary comparisons with more aggregation strategies, as well as the results on the computational runtime of each method.

## I.1    RESULTS WITH DIVERSE DISTRIBUTIONS

In the synthetic data experiments under the *Covariance* setting, we assume the input noise follows a Gaussian distribution. In the following, we further examine the results under alternative noise distributions. Specifically, we fix the number of clients to $K = 3$, with their sample sizes following the ratio $n_1 : n_2 : n_3 = 1 : 1 : 2$. We vary the sample size of Client 1 with

$n_1 \in \{100, 150, 200, 250, 300, 400\}$. We conduct 100 independent trials and report the average results. For the alternative hypothesis $\mathcal{H}_1$, the data are generated as follows:

- **Client 1:** $X \sim \text{Distribution}(\cdot)$, $\quad Y = 0.5X + \epsilon$, $\epsilon \sim \text{Distribution}(\cdot)$;
- **Client 2:** $X \sim \text{Distribution}(\cdot)$, $\quad Y = -0.5X + \epsilon$, $\epsilon \sim \text{Distribution}(\cdot)$;
- **Client 3:** $X \sim \text{Distribution}(\cdot)$, $\quad Y = 0.02X + \epsilon$, $\epsilon \sim \text{Distribution}(\cdot)$.

For the null hypothesis $\mathcal{H}_0$, the data are generated as follows:

- **Client 1:** $X \sim \text{Distribution}(\cdot)$, $\quad Y \sim \text{Distribution}(\cdot)$;
- **Client 2:** $X \sim \text{Distribution}(\cdot)$, $\quad Y \sim \text{Distribution}(\cdot)$;
- **Client 3:** $X \sim \text{Distribution}(\cdot)$, $\quad Y \sim \text{Distribution}(\cdot)$.

Here, $\text{Distribution}(\cdot)$ is drawn from $\{\text{Laplace}(0, 1), \text{Uniform}(-2, 2)\}$.

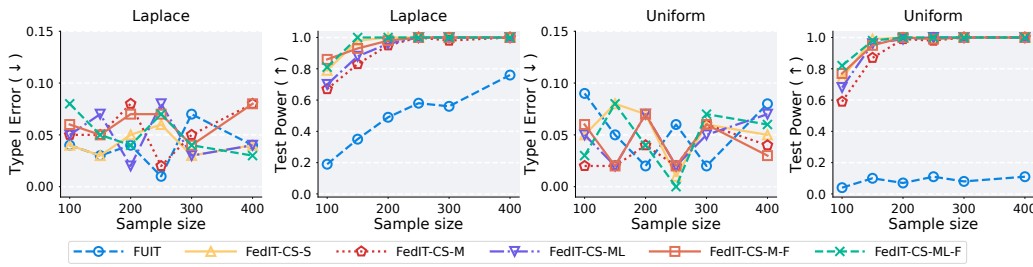

Figure 7: Results under Laplace and Uniform Distributions.

**Performance and analysis.** The experimental results are presented in Fig. 7. All methods successfully control the Type I error rate across various distributional settings and sample sizes. Compared with FUIT, our methods—FedIT-CS-S, FedIT-CS-M, FedIT-CS-ML, FedIT-CS-M-F, and FedIT-CS-ML-F—demonstrate strong performance under both Laplace and Uniform settings, thereby confirming the effectiveness of our stacking-based aggregation strategy. Furthermore, when comparing the "-F" and non-"F" variants, we observe a noticeable difference in detection power, highlighting an important direction for future improvement. Finally, although FedIT-CS-ML was originally designed as an accelerated version of FedIT-CS-M, the additional optimization space effectively leads to a softer solution, which in turn further enhances its detection capability.

## I.2 RESULTS ACROSS FUNCTIONAL RELATIONSHIPS AND CLIENT SCALES

In the above experiments, the number of clients is fixed at three, and the functional relationships within each client are relatively simple, typically defined by predetermined coefficients. In the following, we extend our study with additional experiments under varying functional relationships and client scales. We consider two settings: linear and nonlinear.

***Linear case.*** For the alternative hypothesis $\mathcal{H}_1$, the data are generated as follows:

$$X \sim \mathcal{N}(0, 1), \quad Y = aX + \epsilon_0, \ \epsilon_0 \sim \mathcal{N}(0, 1). \tag{33}$$

For the null hypothesis $\mathcal{H}_0$, the data are generated as follows:

$$X \sim \mathcal{N}(0, 1), \quad Y = a\epsilon_1 + \epsilon_2, \ \epsilon_1, \epsilon_2 \sim \mathcal{N}(0, 1). \tag{34}$$

where $a \sim \text{Uniform}(-0.5, 0.5)$ is a random slope parameter.

***Non-linear case.*** For the alternative hypothesis $\mathcal{H}_1$, the data are generated as follows:

$$X \sim \mathcal{N}(0, 1), \quad Y = f(X + \epsilon_0) + \epsilon_1, \ \epsilon_0, \epsilon_1 \sim \mathcal{N}(0, 1). \tag{35}$$

For the null hypothesis $\mathcal{H}_0$, the data are generated as follows:

$$X = f(\epsilon_2), \quad Y = f(\epsilon_3) + \epsilon_4, \ \epsilon_2, \epsilon_3, \epsilon_4 \sim \mathcal{N}(0, 1). \tag{36}$$

where $f(\cdot)$ is random chosen from the set $\{\sin(\cdot), \cos(\cdot), \tanh(\cdot), \exp(-|\cdot|), (\cdot)^2\}$. are independent noise terms. This construction induces heterogeneous functional relationships across clients.

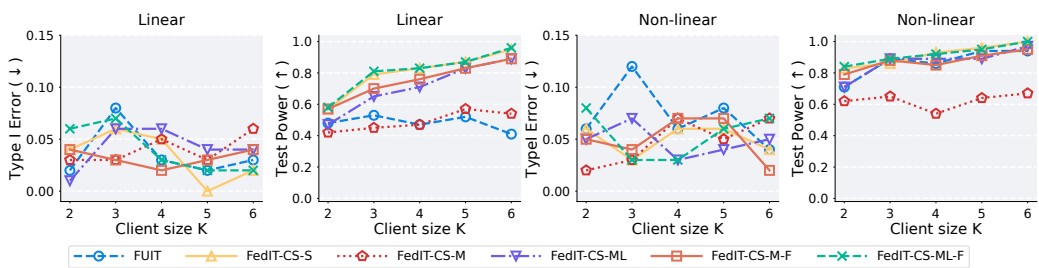

Figure 8: Results across functional relationships (Linear vs. Non-linear) and client scales.

To study the effect of client size, we fix the sample size per client to $n = 200$ and vary the number of clients as $K \in \{2, 3, 4, 5, 6\}$. For each configuration, we conduct 100 independent trials and report the average results. The experimental results are presented in Fig. 8.

**Results of Linear case.** In the linear setting, all methods successfully control the Type I error rate. Compared with FUIT and FedIT-CS-M, FedIT-CS-ML exhibits a clear advantage, demonstrating the effectiveness of its selection strategy. While the performance of FedIT-CS-M is comparable to FUIT, it gradually shows superiority as $K$ increases. When comparing the "-F" and non-"F" variants, we observe a noticeable difference in detection power, highlighting an important direction for future improvement. This gap is particularly pronounced for FedIT-CS-M, likely because its selection component suffers from limited sample size and therefore fails to provide additional benefits. Moreover, FUIT does not gain from the increased sample size as $K$ grows, which further corroborates our theoretical claim that its naive aggregation strategy leads to dependency dilution.

**Results of the Nonlinear case.** Except for FUIT under the nonlinear setting with $K = 3$, all methods successfully control the Type I error rate. Regarding detection power, FedIT-CS-S achieves the best performance in this setting, while FedIT-CS-ML and FUIT also perform well. In contrast, FedIT-CS-M shows inferior performance; comparing it with FedIT-CS-M-F suggests that sample splitting leads to a loss of power. Since our split ratio is set to $0.2$, the selection component becomes ineffective with the resulting small training sample size (only $100 \times 0.2 = 20$ for each client), which explains the degraded performance. By contrast, FedIT-CS-ML, though based on the same sample size, still achieves strong results.

## I.3 COMPARISON WITH MORE AGGREGATION STRATEGIES

In Sec. 4.3, we claimed that using only the aggregated coefficients for the selection step already yields satisfactory performance. In this section, we empirically validate this claim by comparing it against a permutation-based alternative, named FedIT-CS-MB. Specifically, FedIT-CS-MB is derived from FedIT-CS-M by replacing the correlation-based modeling of power with a permutation-based approach, where each client transmits $B+1$ sets of statistics to the server, from which $p$-values are estimated to approximate the method's power. We compare FedIT-CS-MB and FedIT-CS-M from two perspectives: performance and efficiency. For the error rate comparison, the experimental setup follows the covariance setting in Sec. H.1 with $n_1 = 150$. For the runtime comparison, we adopt the linear case described in Sec. I.2. In each experiment, the per-client sample size is fixed at $n = 200$, while the number of clients is varied as $K \in \{2, 4, 8, 16\}$ to evaluate scalability.

**Performance and analysis.** The experimental results are shown in the left two panels of Fig. 9. The leftmost panel compares the two methods in terms of Type I and Type II error rates. Both methods successfully control the Type I error rate; however, FedIT-CS-M achieves a lower Type II error rate than FedIT-CS-MB. This indicates that the correlation-coefficient-based aggregation criterion is already able to capture sufficient dependency signals, even without resorting to permutation-based modeling. Turning to the middle panel, note that both methods have a theoretical complexity of $O(2^K)$, but permutation introduces an additional factor $B$, which directly affects scalability. As shown, FedIT-CS-MB can only handle up to 8 clients within roughly 500 seconds, whereas FedIT-CS-M is able to process 16 clients in about 20 seconds. Overall, the aggregation strategy of FedIT-CS-M has been proved to be more effective than the permutation-based approach, achieving faster computation while maintaining competitive accuracy.

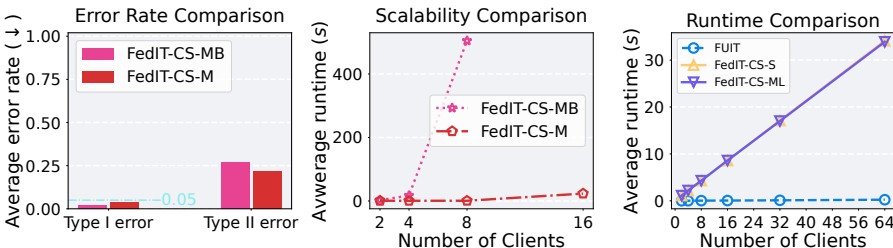

Figure 9: Left and Middle: The performance comparison between FedIT-CS-MB and FedIT-CS-M. Right: The average runtime results of large client set.

### I.4 RUNTIME EVALUATION

In this section, we also provide a runtime comparison of the methods with linear-time complexity with respect to the number of clients, namely FUIT, FedIT-CS-S, and FedIT-CS-ML. For the runtime evaluation, we adopt the linear case described in Sec. I.2. In each experiment, the per-client sample size is fixed at $n = 200$, while the number of clients is varied as $K \in \{2, 4, 8, 16, 32, 64\}$.

**Performance and analysis.** The experimental results are presented in Fig. 9, shown in the rightmost panel. All three methods exhibit linear-time complexity with respect to the number of clients and can thus scale to large client number cases (e.g., $K = 64$). FUIT shows relatively competitive runtime performance. By contrast, FedIT-CS-S and FedIT-CS-ML require permutation-based testing to obtain $p$-values with theoretical guarantees, which introduces an additional computational factor of approximately $B = 100$. We adopt the permutation-based approach because of its strong empirical performance and its applicability to arbitrary input distributions. In the future, non-permutation alternatives, such as exact distributional estimation methods, may be developed to further improve efficiency, though their method design remains technically challenging at present.

**Remark.** From a practical perspective, although permutation incurs extra cost, the intra-client computations across the $B$ samples are mutually independent and can therefore be parallelized. Such parallelization effectively mitigates the constant overhead, rendering our framework not only theoretically sound but also practically scalable in real-world distributed settings.

## J  LIMITATIONS AND BROADER IMPACTS

**Limitations.** As shown in both the main experiments and the additional results in Appendix I, some variants of our framework rely on data splitting to ensure Type I error control. However, this splitting inevitably reduces statistical power. We view this as a key limitation of the current design, and we hope future work will explore more advanced strategies to mitigate this trade-off and further improve the test's effectiveness.

**Broader impacts.** This work proposes a novel framework for federated independence testing. The proposed linear-time aggregation strategy can be optimized in a data-driven manner, ensuring both effectiveness and efficiency. This could be beneficial for developing more reliable downstream algorithms in a variety of areas, including causal discovery, feature selection, and deep learning.

## K  USE OF LARGE LANGUAGE MODELS: AN EXPLANATION

During the preparation of this manuscript, we employed ChatGPT as a writing assistant. Specifically, we provided the prompt: "I am preparing a paper for submission to an international conference and would like your help to check for any grammatical issues and refine the wording or sentence structure where necessary to ensure conciseness and precision." The model's suggestions were applied on a paragraph-by-paragraph basis, and all outputs were carefully reviewed and edited by the authors to ensure accuracy and appropriateness.

