# OpenReview forum: "Powerful Independence Testing on Heterogeneous Federated Clients with Theoretical Guarantees"
_ICLR.cc/2026/Conference — Submitted to ICLR 2026_

### Official Review · Reviewer_q9Rs · 2025-10-19

**Soundness:** 3
**Presentation:** 3
**Contribution:** 2
**Rating:** 6
**Confidence:** 2

**Summary:**

The authors investigate the problem of federated independence testing across heterogeneous local distributions by introducing a copula-based marginal alignment and stacking-aggregation method with privacy-preserving secure aggregation. They provide rigorous theoretical guarantees for Type I error control and statistical power, and validate their approach through both simulated and real-world data experiments.

**Strengths:**

This paper explores an interesting and practical topic — federated independence testing. The authors present a novel method supported by rigorous theoretical guarantees, and importantly, they incorporate privacy provisions that earlier approaches overlooked.

**Weaknesses:**

1. The consistent dependence assumption considered in this paper is quite restrictive, and may not hold in many real-world federated settings where dependence structures differ across domains.

2. The experiments appear relatively simple: the authors focus mainly on small-scale settings (for example, only three participating clients) and use quite basic data-generation processes.

3. The novelty of the proposed method seems incremental: the approach appears to be a direct combination of existing techniques, such as copula transforms, random projection, homomorphic encryption and so on.

**Questions:**

1. Assumption 1 is very strong and restrictive. Could the authors provide more concrete application scenarios that satisfy this assumption? In those cases, what are X and Y (in each silo), and why would the correlation / dependence structure be consistent across silos? Also, it is recommended to cite more literature that supports such settings.
2. Assumption 2 posits that only the marginal distributions are heterogeneous across clients. Could the authors consider the case where the joint distribution ($P_{X,Y}$) or the conditional distribution ($P_{Y|X}$) is heterogeneous (i.e., concept shift), which is also common in federated learning? Does the proposed method apply (or can it be extended) to such scenarios?
3. On lines 170-171: why is $f_k := H_k \phi(x^{k})$ defined this way, but then $C_{xy} = \frac1n \sum_k f_k^{T} f_k$? Could the authors clarify this derivation?
4. At present the proposed method appears to consist of a relatively straightforward combination of existing methods. Could the authors explicitly introduce the specific technical challenges their method addresses?
5. Regarding the experimental section: other than comparing with FUIT, the baselines seem to be variants of the proposed method. Would it be possible to include more state-of-the-art alternatives for comparison? Besides, the scale of experiments is small: only 3 clients, small sample sizes, low dimensionality of variables. Could the authors expand the empirical evaluation to larger number of clients, higher dimensional feature spaces, and more realistic federated settings?

---

> ### Author Response · Authors · 2025-11-17
> **Response to Reviewer q9Rs**
>
> Thanks a lot for your comments and questions. Following are our responses to your comments and answers to your questions.
>
> **(W1, Q1) Consistent dependence assumption**
>
> Note that this assumption is standard in federated causal discovery and federated independence testing [1, 2]. Nonetheless, it can be **relaxed** without breaking our framework:
>
> - Conceptually, the testing goal can be modified to detect whether **any client** exhibits dependence.
> - Theoretically, our guarantees still hold under this relaxation.
> - Practically, our maximization-based aggregation naturally **amplifies strong dependence** rather than averaging it away.
>
> To further validate this, we added a new experiment:
>
> Setup: Covariance setting with 5 clients having correlations [-0.5, 0.5, 0, 0, 0]， (i.e., two dependent, three  independent)
>
> | n₁   | FUIT | FedIT-CS-S | FedIT-CS-ML | FedIT-CS-ML-F |
> | ---- | ---- | ---------- | ----------- | ------------- |
> | 100  | 0.07 | 0.65       | 0.51        | 0.80          |
> | 200  | 0.13 | 0.98       | 1.00        | 1.00          |
> | 300  | 0.23 | 1.00       | 1.00        | 1.00          |
> | 400  | 0.24 | 1.00       | 1.00        | 1.00          |
>
> These results confirm that our method continues to work effectively even when dependence is present in only a subset of clients.
>
> [1] Loka Li, Ignavier Ng, Gongxu Luo, Biwei Huang, Guangyi Chen, Tongliang Liu, Bin Gu, and Kun Zhang. Federated causal discovery from heterogeneous data. arXiv preprint arXiv:2402.13241, 2024.
>
> [2] Biwei Huang, Kun Zhang, Jiji Zhang, Joseph Ramsey, Ruben Sanchez-Romero, Clark Glymour, and Bernhard Scholkopf. Causal discovery from heterogeneous/nonstationary data. ¨ Journal of Machine Learning Research, 21(89):1–53, 2020
>
> **(W3, Q4) Novelty and technical challenges**
>
> We respectfully disagree with the claim that the contribution is incremental. The novelty of our work lies in the following key aspects:
>
> 1. **New Problem Setting**: We tackle *federated independence testing under heterogeneity*—a setting fundamentally different from both standard independence testing and federated causal discovery. Existing methods (e.g., FUIT) do not address the issues of spurious aggregation and dependence dilution that arise under heterogeneous silos.
> 2. **Systematic Design Framework**: Our contribution goes beyond assembling known tools. We propose a *framework-level solution* that strategically integrates copula transforms, nonlinear feature mapping, and customized aggregation to enable:
>    - Encrypted communication with just one-dimensional statistics (crucial for HE efficiency)
>    - Avoidance of dependence dilution, a problem unaddressed in prior works
> 3. **Modularity and Generality**: Each component (e.g., copula for marginal alignment) is replaceable, but the **structural insight**—decoupling marginal alignment, nonlinear modeling, and federated aggregation—is core to solving FedIT under heterogeneity. This abstraction generalizes beyond our specific implementation.
> 4. **Comprehensive Guarantees and Results**: We provide **theoretical Guarantee**and demonstrate **strong empirical performance** across synthetic and real-world settings—*with privacy*, *scalability*, and *accuracy*, a combination not achieved by prior methods.
>
> **(Q2) Joint or conditional heterogeneity (concept shift)**
>
> Assumption 2 is actually more flexible than it may appear: **it does not restrict changes only to marginal distributions**. It also allows shifts in functional relationships between $X$ and $Y$. Therefore, the assumption already covers many forms of heterogeneity, including those resembling changes in $P_{XY}$ and $P_{Y|X}$ (concept shift). Our framework continues to apply because dependence testing — unlike prediction — does not require the conditional model to be shared across clients. We will clarify this in the revision.

---

> ### Author Response · Authors · 2025-11-17
> **Response to Reviewer q9Rs**
>
> Thanks a lot for your comments and questions. Following are our responses to your comments and answers to your questions.
>
> **(Q3) Clarification of derivation on lines 170–171**
>
> The derivation proceeds directly from the definitions: From Line 165, $C\_{xy}^{(k)}=\frac{1}{n}\phi(\mathbf{x}^{k})^T\mathbf{H}\_k^T\mathbf{H}\_k\phi(\mathbf{x}^{k})$ . From Line 170, $C\_{xy}=\sum_k C\_{xy}^{(k)}$ . To expose the limitation of the original design, we define the actual aggregated feature per client: $f\_k=\mathbf{H}\_k\phi(\mathbf{x}^{k})$. Substituting this into the expression above yields: $C\_{xy}^{(k)}=\frac{1}{n}f\_k^Tf\_k$, thus $C\_{xy} = \sum\_k C\_{xy}^{(k)}=\frac{1}{n}\sum\_k f\_k^Tf\_k$ which matches the form stated in the paper.
>
> **(W2， Q5) Additional baselines and expanded evaluation**
>
> ##### **(1) Larger-scale federated settings (more clients, larger samples)**
>
> Following the suggestion, we now include experiments with **20–50 clients**, where each client holds 50 samples. Results show that FedIT maintains high power even as the number of clients grows, the setting is following linear case (line 1500) except the client number in our main paper:
>
> | #Clients (K) | Total n | FUIT | FedIT-CS-S | FedIT-CS-ML |
> | ------------ | ------- | ---- | ---------- | ----------- |
> | 10           | 500     | 0.18 | 0.17       | 0.13        |
> | 20           | 1000    | 0.26 | 0.39       | 0.30        |
> | 30           | 1500    | 0.30 | 0.61       | 0.51        |
> | 40           | 2000    | 0.28 | 0.72       | 0.55        |
> | 50           | 2500    | 0.31 | 0.83       | 0.64        |
>
> These trends confirm that our method scales well to realistic federated scenarios with many clients.
>
> ##### **(2) More baselines**
>
> We acknowledge that federated independence tests are scarce due to the novelty of the setting. FUIT — originally designed for federated causal discovery — remains the closest and strongest baseline.
>
> However, motivated by the reviewer’s suggestion, we additionally adapted **state-of-the-art centralized independence tests**: HSIC-Agg [3], dCor [4], by pooling client samples during aggregation. We note that this approach: does not preserve privacy,  incurs high communication cost.
>
> Below are results under the Cov setting:
>
> | n₁      | FUIT | FedIT-CS-S | FedIT-CS-M | FedIT-CS-ML | FedIT-CS-M-F | FedIT-CS-ML-F | HSIC-Agg | dCor |
> | ------- | ---- | ---------- | ---------- | ----------- | ------------ | ------------- | -------- | ---- |
> | **100** | 0.14 | 0.77       | 0.66       | 0.63        | 0.79         | 0.83          | 0.12     | 0.02 |
> | **150** | 0.14 | 0.99       | 0.78       | 0.89        | 0.94         | 0.99          | 0.55     | 0.07 |
> | **200** | 0.19 | 1.00       | 0.95       | 0.98        | 0.98         | 1.00          | 0.73     | 0.11 |
> | **250** | 0.11 | 1.00       | 0.97       | 1.00        | 1.00         | 1.00          | 0.91     | 0.13 |
> | **300** | 0.24 | 1.00       | 1.00       | 1.00        | 1.00         | 1.00          | 0.98     | 0.13 |
> | **400** | 0.29 | 1.00       | 1.00       | 1.00        | 1.00         | 1.00          | 1.00     | 0.10 |
>
> As seen, HSIC-Agg and dCor suffers from dependency dilution thus not acheive best performance.
>
> ##### **(3) Higher-dimensional setting**
>
> We further evaluate under a 4-dimensional case following [5], with client-specific rotations
>  $\theta  \in [0.2\*π/4, 0.3\*π/4, 0.4\*π/4]$, creating heterogeneous dependency strengths:
>
> | n₁      | HSIC-Agg | dCor | FUIT | FedIT-CS-S | FedIT-CS-M | FedIT-CS-ML | FedIT-CS-M-F | FedIT-CS-ML-F |
> | ------- | -------- | ---- | ---- | ---------- | ---------- | ----------- | ------------ | ------------- |
> | **300** | 0.17     | 0.07 | 0.08 | 0.35       | 0.19       | 0.35        | 0.31         | 0.45          |
> | **450** | 0.45     | 0.05 | 0.06 | 0.64       | 0.42       | 0.62        | 0.57         | 0.73          |
> | **600** | 0.72     | 0.11 | 0.09 | 0.86       | 0.63       | 0.72        | 0.62         | 0.83          |
>
> These results demonstrate that our methods still perform well.
>
>
> [3] Schrab, A., Kim, I., Guedj, B., and Gretton, A. (2022). Efficient aggregated kernel tests using incomplete
> u-statistics. Advances in Neural Information Processing Systems, 35:18793–18807.
>
> [4] Székely, G., Rizzo, M., and Bakirov, N. (2007). Measuring and testing dependence by correlation of
> distances. Annals of Statistics, 35(6):2769–2794.
>
> [5] Gretton, A., Fukumizu, K., Teo, C., Song, L., Schölkopf, B., and Smola, A. (2007). A kernel statistical test
> of independence. Advances in neural information processing systems, 20.

---

### Official Review · Reviewer_gdKX · 2025-10-30

**Soundness:** 3
**Presentation:** 3
**Contribution:** 2
**Rating:** 4
**Confidence:** 3

**Summary:**

This paper tackles the challenge of testing statistical independence in federated learning systems where data distributions vary significantly across clients. It introduces a framework called FedIT CS that aligns client data distributions using copula transforms and aggregates dependency signals through a stacking based mechanism. The method provides theoretical guarantees for controlling Type I error while maintaining detection power, even under client heterogeneity.
Extensive experiments demonstrate its superiority over existing approaches in both synthetic and real world datasets.

**Strengths:**

S1) This paper points out and demonstrates that FUIT's aggregation strategy is equivalent to the naive concatenation of samples in the feature space, and theoretically explains why such a simplistic approach fails in heterogeneous settings.
S2）The proposed method strikes a balance between privacy and efficiency. FedIT-CS-ML significantly reduces computational complexity while maintaining performance.
S3）This theorem provides a rigorous guarantee that the proposed federated independence test strictly controls the Type I error rate at or below the nominal significance level α.

**Weaknesses:**

W1) The core components employed in this paper, including Copula, RFF+CCA, permutation testing and homomorphic encryption, all represent existing methodologies. The paper's principal contribution resides in their systematic integration and domain-specific adaptation rather than pursuing  theoretical breakthroughs.
W2) The paper relies solely on the Sachs dataset for real world validation. This biological network benchmark contains a limited number of variables with well defined relationships and high data quality, which fails to represent the more chaotic and complex scenarios commonly encountered in practical federated learning environments. Consequently, the method's success on this curated dataset cannot convincingly demonstrate its generalizability. Additional experiments on more diverse real world datasets would be necessary to strengthen the persuasiveness of the proposed approach.
W3) The method works on the assumption that all clients must show the same kind of relationship all together—either all independent or all dependent. But in real situations, if some clients show independence while others show dependence, the goal of this method becomes unclear, and the way it combines results gets confusing. As a result, the method's promises might not hold. For example, Hospital A mainly treats patients with serious diabetes. There, insulin levels and blood sugar values are strongly connected. On the other hand, a health check-up center with mostly healthy people may show no clear connection that the method can find between these two measures. This is because a healthy body keeps them in balance naturally.

**Questions:**

Please refer to W1-W3.

---

> ### Author Response · Authors · 2025-11-17
> **Response to Reviewer gdKX (W1 & W3)**
>
> Thank you very much for your comments! Please check our responses are as follows:
>
> **(W1) Novelty**
>
> We respectfully disagree with the reviewer’s assertion. Although our approach incorporates established techniques such as copula transforms, RFF+CCA, permutation testing, and Homomorphic Encryption (HE), the contribution is **not a simple combination of existing tools**. The novelty lies in how these components are **systematically integrated and theoretically grounded** to address the unique challenges of **federated independence testing under heterogeneity**, an area not adequately covered by existing methods.
>
> Our main contributions are as follows:
>
> 1. **New Problem Setting**: We introduce and systematically analyze **FedIT under heterogeneity**, a setting where conventional methods suffer from spurious aggregation and dependence dilution—limitations not addressed by existing federated causal discovery approaches such as FUIT.
> 2. **Theoretical Advances**: We prove that **marginal alignment using copula transforms**, together with nonlinear feature mappings, enables valid aggregation under heterogeneous marginal distributions. Moreover, we provide **Type I error control guarantees**, filling a theoretical gap in prior work.
> 3. **Framework-Level Innovation**: Instead of merely assembling known components, we propose a **modular framework** that strategically separates marginal alignment, nonlinear embedding, and privacy-preserving aggregation. This design:
>    - Reduces encrypted communication to **single-dimensional statistics**, making HE practical in federated settings.
>    - Alleviates **dependence dilution**, a key challenge in heterogeneity-aware aggregation.
> 4. **Modularity and Generality**: Although we use copula transforms in our implementation, the alignment module is replaceable—one could substitute alternative distribution alignment methods while preserving theoretical properties.
> 5. **Robust Empirical Results**: The framework is comprehensively validated across synthetic and real-world scenarios, demonstrating a rare combination of **high power**, **scalability**, and **privacy preservation**, which is absent in current federated independence testing solutions.
>
> **(W3) Assumption that all clients share the same dependence status**
>
> Fisrt of all, we should point out that this assumption is standard in federated causal discovery and federated independence testing [1, 3]. Nonetheless, it can be **relaxed** without breaking our framework:
>
> - Conceptually, the testing goal can be modified to detect whether **any client** exhibits dependence.
> - Theoretically, our guarantees still hold under this relaxation.
> - Practically, our maximization-based aggregation naturally **amplifies strong dependence** rather than averaging it away.
>
> In the reviewer’s example — where one hospital sees severe diabetic cases (strong insulin–glucose dependence) and another sees mostly healthy patients (weak or no dependence) — our method would automatically assign **larger weights to the dependent clients**. These learned weights are also **interpretable**, revealing which clients contribute most to the detected relationship.
>
> To further validate this, we added a new experiment:
>
> Setup: Covariance setting with 5 clients having correlations [-0.5, 0.5, 0, 0, 0]， (i.e., two dependent, three  independent)
>
> | n₁   | FUIT | FedIT-CS-S | FedIT-CS-ML | FedIT-CS-ML-F |
> | ---- | ---- | ---------- | ----------- | ------------- |
> | 100  | 0.07 | 0.65       | 0.51        | 0.80         |
> | 200  | 0.13 | 0.98       | 1.00        | 1.00          |
> | 300  | 0.23 | 1.00       | 1.00        | 1.00          |
> | 400  | 0.24 | 1.00       | 1.00        | 1.00          |
>
> These results confirm that our method continues to work effectively even when dependence is present in only a subset of clients.
>
> [1] Loka Li, Ignavier Ng, Gongxu Luo, Biwei Huang, Guangyi Chen, Tongliang Liu, Bin Gu, and Kun Zhang. Federated causal discovery from heterogeneous data. arXiv preprint arXiv:2402.13241, 2024.
>
> [2] C. M. Bird and N. Burgess, “The hippocampus and memory: insights from spatial processing,” *Nature Reviews Neuroscience*, vol. 9, no. 3, pp. 182–194, 2008.
>
> [3] Biwei Huang, Kun Zhang, Jiji Zhang, Joseph Ramsey, Ruben Sanchez-Romero, Clark Glymour, and Bernhard Scholkopf. Causal discovery from heterogeneous/nonstationary data. ¨ Journal of Machine Learning Research, 21(89):1–53, 2020

---

> ### Author Response · Authors · 2025-11-17
> **Response to Reviewer gdKX (W2)**
>
> Thank you very much for your comments! Please check our responses are as follows:
>
> **(W2) More real-world validation**
>
> We appreciate the reviewer’s feedback. To strengthen the empirical rigor beyond the Sachs dataset, we have added a more complex real-world experiment using an fMRI hippocampus dataset, which has been widely used in federated causal discovery research (e.g., [1]). The dataset records activity from six brain regions (e.g., PRC, PHC, ERC, Sub, CA1, CA3), across 84 consecutive days, which we treat as **84 federated clients**. This setting allows us to assess our method under **larger and more realistic federated configurations**.
>
> Importantly, this dataset exhibits strong temporal dependence, a phenomenon common in real-world domains such as finance. As shown by our ACF analysis:
>
> - Lag 1 ≈ 0.95 → Strong short-term dependence (typical AR(1))
> - Lag 2–3 decay exponentially → Suggestive of ARMA(1,1)
> - Long lags near zero → Weak stationarity
>
> According to [2], only PHC and PRC are truly independent, which we use to estimate Type I error. Here, FUIT fails completely (Type I ≈ 0.88), while our FedIT-CS-ML improves control to ≈ 0.53. The poor performance reflects that **naive i.i.d. permutation is invalid under autocorrelation**. To address this, we further  incorporate **circular-shift permutation**, preserving temporal structure.
>
> With this refinement, the Type I error results for the only truly independent pair PHC vs PRC (sampling 10 clients out of 84 per trial) are:
>
> | Method      | Type I Error |
> | ----------- | ------------ |
> | FedIT-CS-S  | 0.11         |
> | FedIT-CS-ML | 0.10         |
>
> This shows that our framework, after a **simple modification** (circular-shift permutation), can be applied to this substantially more complex temporal federated scenario while still maintaining **good Type I error control**.
>
> We further assess **power** on a dependent pair. Following [2], CA1 → Sub implies dependence between CA1 and downstream regions. Focusing on CA1 and Sub and varying the number of clients, we obtain:
>
> | #Clients | FedIT-CS-S | FedIT-CS-ML |
> | -------- | ---------- | ----------- |
> | 5        | 0.78       | 0.66        |
> | 10       | 0.93       | 0.89        |
> | 20       | 1.00       | 0.99        |
>
> As the number of clients increases, both variants show **monotonic improvement in power**, further supporting the effectiveness of our method in this real-world federated setting.
>
> We also extend our **synthetic experiments** to more complex federated configurations. We now include **20–50 clients**, each with 50 samples (same linear setting as in the main paper, with only the client count varied). The results are:
>
> | #Clients (K) | Total n | FUIT | FedIT-CS-S | FedIT-CS-ML |
> | ------------ | ------- | ---- | ---------- | ----------- |
> | 10           | 500     | 0.18 | 0.17       | 0.13        |
> | 20           | 1000    | 0.26 | 0.39       | 0.30        |
> | 30           | 1500    | 0.30 | 0.61       | 0.51        |
> | 40           | 2000    | 0.28 | 0.72       | 0.55        |
> | 50           | 2500    | 0.31 | 0.83       | 0.64        |
>
> These trends confirm that FedIT remains **robust and scalable** as the number of clients grows, both in challenging real-world data and synthetic settings.
>
> [1] Loka Li, Ignavier Ng, Gongxu Luo, Biwei Huang, Guangyi Chen, Tongliang Liu, Bin Gu, and Kun Zhang. Federated causal discovery from heterogeneous data. arXiv preprint arXiv:2402.13241, 2024.
>
> [2] C. M. Bird and N. Burgess, “The hippocampus and memory: insights from spatial processing,” *Nature Reviews Neuroscience*, vol. 9, no. 3, pp. 182–194, 2008.
>
> [3] Biwei Huang, Kun Zhang, Jiji Zhang, Joseph Ramsey, Ruben Sanchez-Romero, Clark Glymour, and Bernhard Scholkopf. Causal discovery from heterogeneous/nonstationary data. ¨ Journal of Machine Learning Research, 21(89):1–53, 2020

---

> ### Author Response · Authors · 2025-11-20
> **Looking forward to your further comments!**
>
> Dear Reviewer gdKX,
>
> Thanks for your comments. We've submitted our responses to your concerns on our work. Would you please check our responses and feedback your further comments?
>
> Thanks again!
>
> Best regards,
>
> The authors

---

### Official Review · Reviewer_VrEb · 2025-11-03

**Soundness:** 3
**Presentation:** 3
**Contribution:** 3
**Rating:** 6
**Confidence:** 3

**Summary:**

This paper addresses the problem of federated independence testing (FedIT) under client heterogeneity, where data distributions (marginal distributions, dependence strength/functional forms) vary across clients, but the global dependence status (independent/dependent) is consistent. It first analyzes limitations of existing methods (e.g., FUIT), which adopt naive feature-space concatenation and lack theoretical guarantees or effective handling of dependence dilution. The paper then proposes the FedIT-CS framework, combining copula-based marginal alignment (to eliminate marginal distribution discrepancies while preserving dependence structures) and stacking-based aggregation (to amplify intra-client dependence signals and select optimal client subsets). The framework includes three variants (FedIT-CS-M, FedIT-CS-ML, FedIT-CS-S) and incorporates homomorphic encryption (HE) for privacy preservation. Theoretically, it proves the soundness of the aggregated statistic (Theorem 4) and Type I error control (Theorem 5). Experimentally, it validates the method on synthetic datasets (covariance, frequency, functional heterogeneity) and the real-world Sachs dataset, showing that FedIT-CS variants outperform FUIT in balancing Type I/II error rates.

**Strengths:**

1.	Systematic problem analysis: The paper clearly identifies two core challenges of FedIT under heterogeneity—"naive aggregation pitfalls" (spurious dependence/independence) and "dependence dilution" (opposing correlations canceling out)—and links these to the limitations of existing methods, providing a clear motivation for the proposed framework.
2.	Theoretical rigor: It establishes formal guarantees for Type I error control and the soundness of the aggregated statistic, filling the gap of theoretical inadequacy in prior FedIT work (e.g., FUIT).
3.	Practical design: The framework balances privacy, efficiency, and performance: HE ensures privacy without accuracy loss; FedIT-CS-ML achieves linear-time complexity (O(KB)) for large-scale clients; and the use of second-order moments reduces communication costs compared to FUIT’s covariance matrix transmission (O(Kh²)).
4.	Comprehensive validation: Experiments cover both synthetic (three heterogeneous scenarios) and real-world datasets, with detailed analysis of Type I/II error rates and scalability, ensuring the method’s robustness across different settings.

**Weaknesses:**

1.	Scalability of FedIT-CS-M: FedIT-CS-M’s exponential complexity (O(2^K B)) makes it infeasible for K > 10 (as shown in Appendix I.3, FedIT-CS-MB—its permutation-based variant—fails to scale beyond 8 clients). The paper does not propose heuristic optimizations (e.g., greedy subset selection) to mitigate this, restricting its use in large federated systems.
2.	Limited real-world validation: The only real-world dataset used is Sachs (a signal network dataset), which has a fixed number of clients (7) and variables (11). No experiments on other domains (e.g., healthcare, finance) or larger client counts (e.g., K=50) are provided, making it hard to assess the framework’s generalizability to diverse real-world FedIT scenarios.
3.	Data splitting trade-off unaddressed: The paper uses a simple 8:2 data split to separate aggregation strategy training and testing, which reduces statistical power (evidenced by lower performance of FedIT-CS-M/FedIT-CS-ML vs. their "-F" variants with extra training data). No alternative strategies (e.g., nested cross-validation, split-free methods like Schrab et al. 2022) are explored, limiting the framework’s adaptability to small-dataset scenarios.
4.	Homomorphic encryption details lacking: Appendix D outlines the HE procedure but does not report key metrics like encryption/decryption time, communication latency, or memory usage. This makes it difficult for practitioners to evaluate the framework’s practicality in low-latency federated environments.

**Questions:**

Please refer to the weaknesses.

---

> ### Author Response · Authors · 2025-11-17
> **Response to Reviewer VrEb (W1 & W2)**
>
> Thanks a lot for your comments. We present our responses as follows:
>
> **(W1) Scalability of FedIT-CS-M.**
>
> We agree that the exponential complexity of FedIT-CS-M limits its scalability when the number of clients grows. This challenge directly motivated our development of the **linear-time variant FedIT-CS-ML**.
>
> Following your comments, we implemented a **greedy subset-selection heuristic**, which iterates through clients once and adds a client only if it increases the aggregated objective. Under the Cov setting (n₁ = 200), we compare this variant (denoted as *Greedy*) against FedIT-CS-M and FedIT-CS-ML, the results are:
>
> | Method      | Power |
> | ----------- | ----- |
> | FedIT-CS-M  | 0.95  |
> | Greedy      | 0.96  |
> | FedIT-CS-ML | 0.99  |
>
> These results demonstrate that the greedy heuristic is effective, achieving performance comparable to FedIT-CS-M while avoiding exponential complexity. In contrast, FedIT-CS-ML further leverages optimization, thus yielding superior results overall.
>
> **(W2) Real-world validation.**
>
> We thank the reviewer for this valuable feedback. We have extended our evaluation with a **new real-world experiment** using an fMRI hippocampus dataset, which has been widely used in federated causal discovery research (e.g., [1]). The dataset records activity from **six brain regions** (e.g., PRC, PHC, ERC, Sub, CA1, CA3), across 84 consecutive days, which we treat as **84 federated clients**. This setting allows us to assess our method under **larger and more realistic federated configurations**.
>
> Importantly, this dataset exhibits strong temporal dependence, a phenomenon common in real-world domains such as finance. As shown by our ACF analysis:
>
> - Lag 1 ≈ 0.95 → Strong short-term dependence (typical AR(1))
> - Lag 2–3 decay exponentially → Suggestive of ARMA(1,1)
> - Long lags near zero → Weak stationarity
>
> According to [2], only PHC and PRC are truly independent, which we use to estimate Type I error. Here, FUIT fails completely (Type I ≈ 0.88), while our FedIT-CS-ML improves control to ≈ 0.53. The poor performance reflects that **naive i.i.d. permutation is invalid under autocorrelation**. To address this, we further  incorporate **circular-shift permutation**, preserving temporal structure.
>
> With this refinement, the Type I error results for the only truly independent pair PHC vs PRC (sampling 10 clients out of 84 per trial) are:
>
> | Method      | Type I Error |
> | ----------- | ------------ |
> | FedIT-CS-S  | 0.11         |
> | FedIT-CS-ML | 0.10         |
>
> This shows that our framework, after a **simple modification** (circular-shift permutation), can be applied to this substantially more complex temporal federated scenario while still maintaining **good Type I error control**.
>
> We further assess **power** on a dependent pair. Following [2], CA1 → Sub implies dependence between CA1 and downstream regions. Focusing on CA1 and Sub and varying the number of clients, we obtain:
>
> | #Clients | FedIT-CS-S | FedIT-CS-ML |
> | -------- | ---------- | ----------- |
> | 5        | 0.78       | 0.66        |
> | 10       | 0.93       | 0.89        |
> | 20       | 1.00       | 0.99        |
>
> As the number of clients increases, both variants show **monotonic improvement in power**, further supporting the effectiveness of our method in this real-world federated setting.
>
> We also extend our **synthetic experiments** to larger federated configurations. Following the reviewer’s suggestion, we now include **20–50 clients**, each with 50 samples (same linear setting as in the main paper, with only the client count varied). The results are:
>
> | #Clients (K) | Total n | FUIT | FedIT-CS-S | FedIT-CS-ML |
> | ------------ | ------- | ---- | ---------- | ----------- |
> | 10           | 500     | 0.18 | 0.17       | 0.13        |
> | 20           | 1000    | 0.26 | 0.39       | 0.30        |
> | 30           | 1500    | 0.30 | 0.61       | 0.51        |
> | 40           | 2000    | 0.28 | 0.72       | 0.55        |
> | 50           | 2500    | 0.31 | 0.83       | 0.64        |
>
> These trends confirm that FedIT remains **robust and scalable** as the number of clients grows, both in challenging real-world data and controlled synthetic settings.
>
> [1] Loka Li, Ignavier Ng, Gongxu Luo, Biwei Huang, Guangyi Chen, Tongliang Liu, Bin Gu, and Kun Zhang. Federated causal discovery from heterogeneous data. arXiv preprint arXiv:2402.13241, 2024.
>
> [2] C. M. Bird and N. Burgess, “The hippocampus and memory: insights from spatial processing,” *Nature Reviews Neuroscience*, vol. 9, no. 3, pp. 182–194, 2008.

---

> ### Author Response · Authors · 2025-11-17
> **Response to Reviewer VrEb (W3 & W4)**
>
> Thanks a lot for your comments. We present our responses as follows:
>
> **(W3) Data-splitting trade-off.**
>
> We agree that the 8:2 data split may reduce statistical power, as reflected in the slightly lower performance of FedIT-CS-M and FedIT-CS-ML relative to their “-F” variants. However, even under this basic splitting scheme, our methods already significantly outperform the existing baseline (FUIT) across diverse settings. While more advanced split-free strategies (e.g., Schrab et al., 2022) could further enhance performance, our current work focuses on presenting a unified formulation with theoretical guarantees and practical robustness under heterogeneity. We consider extending the split strategy an important direction for future research, as we highlight in Line 365 of the main paper.
>
>
> **(W4) Homomorphic encryption details**
>
> We appreciate the reviewer’s feedback. We clarify that Homomorphic Encryption (HE) is applied **only to a one-dimensional aggregated statistic**, significantly reducing computational and communication overhead — especially compared to methods such as FUIT, which require exchanging full matrices across clients. Below, we provide performance metrics for the HE component. Using the TenSEAL library, each encryption-decryption cycle takes approximately **4 ms per client**, with a communication cost of roughly **80 KB**. The memory footprint is negligible, as confirmed using `psutil`. Thanks to this lightweight design, the cryptographic protection incurs minimal overhead, making HE both efficient and practical for low-latency federated environments.

---

### Official Review · Reviewer_st3t · 2025-11-04

**Soundness:** 3
**Presentation:** 3
**Contribution:** 3
**Rating:** 4
**Confidence:** 5

**Summary:**

To tackle independence testing in federated learning, the authors have proposed a copula-based marginal alignment technique combined with a stacking-based aggregation strategy.

**Strengths:**

The paper is overall well written. Type I error bound and soundness of aggregated statistics in Section 5 are the main strengths of this paper.

**Weaknesses:**

My major concern is the lack of comparison with closely related work and baselines. In particular, the independence testing in federated setting can be solved via density ratio estimation and matching:

[1] M. Yamada and M. Sugiyama. Dependence minimizing regression with model selection for non-linear causal inference under non-Gaussian noise. AAAI 2010.

[2] M. Sugiyama and T. Suzuki. Least-squares independence test. IEICE TRANSACTIONS on Information and Systems, 94(6), pp.1333-1336, 2011.


[3] M. Sugiyama, T. Suzuki,  and T. Kanamori. Density-ratio matching under the bregman divergence: a unified framework of density-ratio estimation. Annals of the Institute of Statistical Mathematics, 64(5), pp.1009-1044, 2012.

[4] A. Ramezani-Kebrya, F. Liu, T. Pethick, G. Chrysos, and V. Cevher. Federated learning under covariate shifts with generalization guarantees. TMLR 2023.

[5] Z. Wu, C. Choi, X. Cao, V. Cevher, and A. Ramezani-Kebrya. Addressing label shift in distributed learning via entropy regularization. ICLR 2025.

For protecting privacy, the authors have used Homomorphic Encryption (HE). However, I do not see any novelty compared to typical aggregations with HE.

**Questions:**

Please address the weaknesses.

---

> ### Author Response · Authors · 2025-11-17
> **Response to Reviewer st3t**
>
> We thank Reviewer st3t for her/his comments. Following are our responses.
>
> **(W1) About comparison with the mentioned works [1]–[5].**
>
> First, for works based on density-ratio estimation [1]–[3], we must point out that **none of these methods are designed for federated settings.** Anyway, following your suggestion, we tried our best to adpat these methods to federated setting and compare them with our method. We conducted a thorough search for official or publicly available implementations of these methods. **Unfortunately, no usable codebase exists** — the code links provided in [2] and [3] are no longer accessible. Hence, we had to re-implement the core components of these methods and adapted them to the federated scenario by aggregating the SMI statistics across clients. This baseline is denoted as “SMI” in our experiments. Below, we provide the performance comparison under three federated synthetic settings:
>
> The results of power are reported as follows:
> | Setting       | SMI  | FUIT | FedIT-CS-S | FedIT-CS-ML | FedIT-CS-ML-F |
> | ------------- | ---- | ---- | ---------- | ----------- | ------------- |
> | Cov (n₁=100)  | 0.71 | 0.16 | 0.78       | 0.63        | 0.83          |
> | Freq (n₁=900) | 0.08 | 0.34 | 0.84       | 0.81        | 0.86          |
> | Func (n₁=400) | 0.24 | 0.65 | 0.80       | 0.61        | 0.83          |
>
> These results show that while SMI performs reasonably well in linear cases, it fails to detect dependencies effectively in nonlinear settings, especially under frequency-domain heterogeneity. This is because the SMI statistic is aggregated directly without optimization, thus lacking robustness under heterogeneity — a key distinction between SMI and our proposed Federated Independence Testing framework.
>
> Second, regarding [4]–[5], these works target **federated risk minimization under distribution shifts** such as covariate or label shift. In contrast, our work addresses **federated independence testing**, which involves hypothesis testing between two variables **without assuming specific forms of heterogeneity** — making our framework more general. So these works address a different task from ours. We don't think it is necessary to compare with them. Anyway, we will cite these papers and further clarify the conceptual differences in the revised version.
>
> **(W2) About the novelty of usinge Homomorphic Encryption (HE) in our work.**
>
> We agree that HE itself is not novel. However, we must emphasize that **our framework facilitates HE in a meaningful and efficient way**. Without our nonlinear transformation and structured dependence modeling steps, HE cannot be directly applied since it supports only addition and multiplication, and cannot handle general nonlinear computation.
>
> Moreover, regardless of the dimensionality of the original variables, our method outputs a **one-dimensional statistic before encryption**, which substantially reduces the computational and communication overhead of HE. This greatly reduces the communication and computation overheads associated with HE, making secure aggregation feasible in federated independence testing.

---

> ### Author Response · Authors · 2025-11-20
> **Looking forward to your further comments!**
>
> Dear Reviewer st3t,
>
> Thanks for your comments. We've submitted our responses to your concerns on our work. Would you please check our responses and feedback your further comments?
>
> Thanks again!
>
> Best regards,
>
> The authors

---

### Author Response · Authors · 2025-12-01
**Summary of the Discussion for the Area Chair**

## Summary of the Discussion for the Area Chair

We thank all the reviewers for their time and comments. Unfortunately, although we submitted all responses to the reviewers' comments on Nov. 17, and even sent messages to the reviewers to remind them for discussion and further feedback on Nov. 20, we didn't get any feedback from the reviewers on our responses to their comments up to Nov. 27 --- the day the Openreview "bug" was exposed. We're definitely sure that **our responses have comprehensively addressed all the reviewers' concerns and answered their questions**. We sincerely appreciate the Area Chair’s time and effort in examining the reviewers' comments and our responses for reaching a convincing and fair decision on our work. To facilitate the Area Chair's evaluation and making decision, we **summarize the key points of our responses to the reviewers' comments** as follows:

### For the Reviewer st3t (Rating: 4):

**Reviewer st3t acknowledged the contributions of our work, only mentioned concern on baseline comparisons, which was completely addressed in our responses**. Specifically, the reviewer recognized the **soundness, clarity, and contribution (all rated “good”)**. Her/his only concern was the so-called "lack of comparisons with closely related work and baselines", and the reviewer also listed five specific existing works. Acutally, the first three papers focus on independence testing in centralized settings rather than federated environments, and thus do not address the heterogeneous multi-client challenges inherent to our problem. The remaining two works concern federated risk minimization and rely on stronger assumptions such as covariance shift or label shift; their problem formulation is fundamentally different from federated independence testing and therefore not directly comparable. Therefore, the five references mentioned by Reviewer st3t are acutally NOT really related to our work. However, to respond to the reviewer's concern, we still provided comprehensive comparison results and analyses with these works mentioned by the reviewer. And the experimental results demonstrate the superiority of our work. Unfortunatley, the reviewer didn't have any feedback on our responses up to Nov. 28.

### For the Reviewer VrEb (Rating: 6):

**Reviewer VrEb appreciated our systematic problem formulation, theoretical rigor, practical design, and strong empirical performance.** To address her/his concerns about scalability, real-world applicability, and HE metrics, we supplied additional experiments and clarifications, which all validate the advantage of our work.

### For Reviewer gdKX (Rating: 4):

**Reviewer gdKX acknowledged our motivation, applicability, and theoretical correctness, but her/his concern regarding limited novelty seems a misunderstanding of our key contributions.** Notably, both Reviewer VrEb and Reviewer q9Rs recognized distinct innovative aspects of our method — including the new heterogeneous FedIT problem setting, the rigorous theoretical guarantees, and the privacy-preserving design. In our responses, we explicitly articulated the substantive innovations behind our framework (addressing W1), added a more complex real-world experiment to strengthen empirical validation (addressing W2), and provided mixed-dependence experiments showing that our method remains valid even when clients do not share the same dependence status (addressing W3). We believe these clarifications fully resolve the concerns raised. Unfortunately, we haven't gotten the reviewer's feedabck before Nov. 28.

### For Reviewer q9Rs (Rating: 6):

**Reviewer q9Rs acknowledged the novelty and theoretical rigor of our method**, and **her/his concerns are about additional baselines and assumption generality, which have been addressed** in our rebuttal. Specifically, the reviewer highlighted the innovative problem setting, the strength of our theoretical guarantees, and the inclusion of privacy mechanisms absent in prior work. Their main concerns focused on the need for more baseline comparisons and clarification of the generality of our assumptions. In response, we provided additional baseline experiments and clarified that our method applies more broadly than the simplified setting presented in the paper, further supporting this with new experimental evidence.

---

### Meta-Review · Area_Chair_ATh9 · 2026-01-04

**Summary:**

The manuscript proposes a federated independence testing framework that addresses both theoretical and practical challenges arising from client heterogeneity. The main concerns from the reviewers are:
* comparison with closely related work and baselines;
* scalability;
* limited real-world validation;
* insufficient experimental settings;
* homomorphic encryption details lacking;
* systematic integration and domain-specific adaptation;
* strong assumption: the method works on the assumption that all clients must show the same kind of relationship all together—either all independent or all dependent;
* extension to general cases (e.g., heterogeneous conditional/joint distribution)

After checking all rebuttals, AC believes the manuscript still needs a significant revision, in terms of adding more related work discussion, comparing with more baselines on more realistic scenarios.

**Reviewer Concerns:**

Reviewer st3t questioned the lack of comparison with closely related work and baselines. Though the authors provided some additional results, the AC believes that the more comprehensive experimental results should be provided. Besides, more related work should be discussed, e.g., "On secure distributed hypothesis testing. ISIT 2015".

Reviewer VrEb questioned the scalability, limited real-world validation, insufficient experimental settings, and homomorphic encryption details. The provided additional results, in terms of scalability, real-world validation, and data-splitting choice, cannot convince AC.

Reviewer gdKX and Reviewer q9Rs share a common concern regarding the methodology's novelty, as well as the real-world validation. The AC would encourage the authors to identify the unique challenges when integrating methods from different fields and use more comprehensive numerical results to justify the claims.

**Reviewer Scores:**

Reviewer st3t and Reviewer VrEb may not increase their scores, based on the AC's assessment.

---

### Decision · Program_Chairs · 2026-01-26

Reject